# A Computational Framework for Modeling Emergence of Color Vision in the Human Brain

**Atsunobu Kotani & Ren Ng**
Department of Electrical Engineering and Computer Sciences
University of California, Berkeley
`{akotani,ren}@berkeley.edu`

## Abstract

It is a mystery how the brain decodes color vision purely from the optic nerve signals it receives, with a core inferential challenge being how it disentangles internal perception with the correct color dimensionality from the unknown encoding properties of the eye. In this paper, we introduce a computational framework for modeling this emergence of human color vision by simulating both the eye and the cortex. Existing research often overlooks how the cortex develops color vision or represents color space internally, assuming that the color dimensionality is known a priori; however, we argue that the visual cortex has the capability and the challenge of inferring the color dimensionality purely from fluctuations in the optic nerve signals. To validate our theory, we introduce a simulation engine for biological eyes based on established vision science and generate optic nerve signals resulting from looking at natural images. Further, we propose a bio-plausible model of cortical learning based on self-supervised prediction of optic nerve signal fluctuations under natural eye motions. We show that this model naturally learns to generate color vision by disentangling retinal invariants from the sensory signals. When the retina contains $N$ types of color photoreceptors, our simulation shows that $N$-dimensional color vision naturally emerges, verified through formal colorimetry. Using this framework, we also present the first simulation work that successfully boosts the color dimensionality, as observed in gene therapy on squirrel monkeys, and demonstrates the possibility of enhancing human color vision from 3D to 4D.

## 1 Introduction

> *"Color is the place where our brain and the universe meet." – Paul Klee*

We experience colors in everyday life so effortlessly that it is easy to take the underlying neural computations for granted. In fact, the sensory signals exiting our eye, called optic nerve signals (ONS), are nothing like our color vision (see Fig. 1 and Supplementary Video 0:30 [1]). For example, ONS are spatially warped, akin to an image taken with a fish-eye lens, due to varying densities of photoreceptor cells in the retina (Curcio et al., 1990). ONS does not come in color either – colors of the scene are spectrally sampled by different types of color sensitive cells (cone cells) in the retina, appearing as a layer of spatial noise in the ONS. Furthermore, other processes, such as lateral inhibition and action potentials, render the image structure barely recognizable, in gradient domain where spatiotemporal "edges" dominate. Now, the question is: *how are we still seeing colors?*

Specifically, this paper introduces a novel computational framework for modeling the emergence of human color vision by simulating the eye and the cortex. For the eye, we present a biophysically accurate implementation of a textbook scientific model of the retinal neural circuitry. For the brain, we hypothesize a low-level, self-supervised learning mechanism in the cortex that operates purely on the optic nerve signal stream. For color representation in the brain, we propose modeling color in the brain as a high-dimensional vector, rather than assuming any specific color dimensionality and show that the correct color dimensionality emerges naturally through the proposed learning.

For eye simulation, our goal is to create a computational engine that takes in any spectral image of the world and outputs the corresponding optic nerve signal stream, based on established vision science.

---

[1]Code, video, dataset, and tutorials are available on our project website: https://matisse.eecs.berkeley.edu.

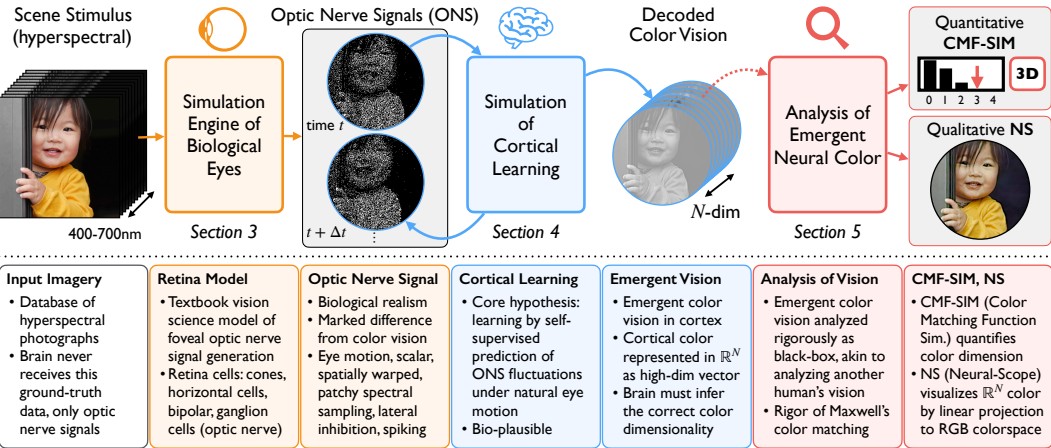

Figure 1: Overview of our proposed framework for modeling the emergence of human color vision. Our simulation engine of biological eyes converts a scene stimulus (hyperspectral image) to a stream of optic nerve signals (Section 3 & Video 0:30). We simulate cortical learning purely from these optic nerve signals (Section 4) and show the emergence of color vision. We show how to analyze the emergent neural color quantitatively with Color Matching Function test Simulator ($CMF\text{-}SIM$) and qualitatively with Neural Scope ($NS$) (Section 5).

Our model captures how scene form, color, and motion become spatiotemporally entangled in the optic nerve signal stream. Our simulation is based on the "textbook" model of vision science (Rodieck, 1998) for midget, private-line visual pathways, detailed in Section 3, and comprising fixational eye motion, spectral encoding by cone cells, foveation, and lateral inhibition.

For brain simulation, we hypothesize that the cortex could disentangle the optic nerve signal from the invariant retinal properties to generate color vision – purely through a self-supervised learning process that aims to predict the constant fluctuations in cellular-level activations of the optic nerve signal during small eye movements. The neural conditions for such self-supervised learning are biologically plausible in the sense that the cortex continuously receives optic nerve signals under the tiny gaze movements of fixational eye drift.The theoretical intuition for why such learning might succeed is that eye motion repeatedly draws a static scene image across the invariant spectral and spatial sampling properties of the retina, potentially enabling the retinal properties and scene images to be mutually filtered out of the optic nerve signal where they are entangled. We show that this simple learning mechanism succeeds at discovering color vision with the correct dimensionality.

But first, a somewhat esoteric yet technically critical feature of the modeling framework needs discussion: color representation in the brain. Existing research often overlooks how the cortex develops color vision or represents color space internally, assuming that the color dimensionality is known a priori, e.g. RGB. We argue that the cortex has the capability and the task of inferring color dimensionality, purely from fluctuations in the optic nerve signals. Therefore, we propose representing color in the cortex as a high-dimensional vector in $\mathbb{R}^N$ and find that the correct color space and color dimensionality emerge naturally as geometric properties of the hypothesized learning mechanism. We show how to formally quantify and visualize the emergent color space (Section 5).

Remarkably, the hypothesized learning mechanism results in clear color vision of the correct color dimensionality. When the retina contains 1, 2, 3 or 4 types of color photoreceptors (cones), correct color dimensionality emerges: respectively, 1D mono-, 2D di-, 3D tri- or 4D tetra-chromatic vision. In fact, it is an esoteric but well-known fact in vision science that the three types of 2D dichromacy result in highly specific color-spaces (blue-yellow, blue-orange, and teal-pink), which we see emerge naturally. Even more, the simulation presents a model of how color dimensionality boosting occurred in the squirrel monkeys (Mancuso et al., 2009). We simulate injection of the third cone pigment virus into the dichromat retina, which results in boosting from dichromacy to trichromacy. Intriguingly, the model also shows the possibility that normal human trichromacy could be boosted to tetrachromacy.

In sum, our proposed framework formulates the emergence of human color vision in a computational manner. The simulation engine of realistic optic nerve signals generates training data to this problem,

and an intentional and challenging constraint is that the cortical model must strictly learn only from the optic nerve signal with no auxiliary information. This paper presents the first, simple and yet complete existence proof that such cortical inference is possible, and we show that this learning simulation is consistent with various surprising and unexplained vision science phenomena.

## 2 RELATED WORK

### 2.1 VISION SCIENCE ON OPTIC NERVE SIGNAL ENCODING

In the "textbook" model, the retina transforms light into electrical signals through three primary functions: color sampling via cone cell spectral response functions, lateral inhibition via horizontal neural connections, and spatial sampling via cell positioning (Fig. 2). Under daylight, this process begins when light activates cone cells, which are of three types for most humans (Young, 1801; 1802; Von Helmholtz, 1867), each sensitive to different wavelengths (Stockman & Sharpe, 2000). These signals are then modulated by horizontal cells that enhance visual contrast through lateral inhibition (Rodieck, 1965; Dacey et al., 1996; Verweij et al., 2003). The signals continue to bipolar and retinal ganglion cells (RGCs), with a direct connection in the fovea via the midget private-line pathway (Dowling & Boycott, 1966; McMahon et al., 2000; Wool et al., 2018), vital for high-resolution color vision. Notably, the cone cell density varies, reaching its peak in the fovea (Osterberg, 1935; Curcio et al., 1990). The axons of the ganglion cells bundle to form optic nerve signals, maintaining their spatial arrangement, which results in a retinotopy in the visual cortex (Holmes, 1918; Tootell et al., 1982; Dougherty et al., 2003). Fixational eye movements cause dynamic photoreceptor activation at all times by constantly shifting the gaze, even when focusing on static objects (Rucci & Victor, 2015; Young & Smithson, 2021; Martinez-Conde et al., 2013).

Recordings of real optic nerve signals exist, but cannot be used for our cortical simulations because they are orders of magnitude too low resolution (from only a few thousand cells) and only in response to grayscale rather than color imagery (Litke et al., 2004; Brackbill et al., 2020; Marre et al., 2017; Liu et al., 2022). To overcome these data limitations, we present a simulation engine for the midget private-line pathway in the fovea in Section 3.

### 2.2 COLOR REPRESENTATION IN COMPUTATIONAL NEUROSCIENCE

It is an open question how to meaningfully model neural representations of color in simulations of visual perception. Most computational neuroscience sidesteps this issue, "hard-coding" a dimensionality of 3, representing and constraining cortical color to tristimulus values, such as RGB (Parthasarathy et al., 2017; Botella-Soler et al., 2018; Brackbill et al., 2020; Kim et al., 2021; Zhang et al., 2022; Wu et al., 2022), LMS (Young, 1802; Von Helmholtz, 1867) or cone-opponent space (Derrington et al., 1984; MacLeod & Boynton, 1979). Constraining cortical models to such a ceiling of three for dimensionality clearly conflicts with the study of a functional tetrachromat observer with 4D color vision (Jordan et al., 2010; Rezeanu et al., 2021). Instead, we model the brain with the capability and the challenge of deducing the inherent color dimensionality encoded in the optic nerve signals.

### 2.3 THEORY ON HOW VISION EMERGES IN THE BRAIN

In computational vision science modeling, it is often overlooked that the cortex relies solely on a stream of optic nerve signals to discover color vision, with no access to a teacher signal or perceptual ground truth. Rather, various efforts have been made to reconstruct visual stimuli from neural responses by giving the cortical model access to ground truth stimulus image (Naselaris et al., 2009; Nishimoto et al., 2011; Parthasarathy et al., 2017; Botella-Soler et al., 2018; Brackbill et al., 2020; Kim et al., 2021), but the neural reality is that the brain never has direct access to the visual scene. This characteristic makes the human visual system a quintessential example of self-supervised learning, which is a growing field in computer vision (De Sa, 1993; Chen et al., 2020; He et al., 2022). In this work, we propose a learning principle in the cortex which aims to predict the cellular-level fluctuations in activation that occur during small eye movements, which is associated with the idea of temporal prediction (Palmer et al., 2015; Lotter et al., 2016; Singer et al., 2018; 2023b), as well as the broader concept of predictive coding (Rao & Ballard, 1999; Srinivasan et al., 1982). It is also closely linked to the sensorimotor contingency theory from cognitive science (O'Regan & Noë, 2001) and

slow feature analysis (Hinton, 1990; Földiák, 1991) which suggests that the brain learns to anticipate the sensory outcomes of motor actions (e.g. eye movements) and filters out invariant (slow) features.

## 2.4 Theory on How Brains Infer Color Dimensionality

Previous studies has investigated inference of invariant retinal properties from sensory signals. For instance, research has demonstrated the ability to deduce cone cell types (Wachtler et al., 2007; Brainard et al., 2008; Benson et al., 2014) and the positions of photoreceptors (Maloney & Ahumada, 1989; Brainard et al., 2008) via statistical methods from sensory signals. Brainard et al. (2008) hinted at analyzing sensory inputs from different time points during fixational drift as a way to reveal retinal features, and in this paper we provide a computational realization of this idea with a specific learning mechanism that achieves complete disentanglement of retinal invariants from optic nerve signals.

## 2.5 Measurement of Color Dimensionality in Human Color Perception

In this paper, we need to rigorously measure the color dimensionality of the emergent color from cortical simulation. To do so, we adapt Maxwell's famous color-matching experiments (Maxwell, 1856), which laid the foundation of colorimetry that remains at the heart of all color reproduction technology today. Color matching experiments confirmed the trichromatic theory (Young, 1802; Von Helmholtz, 1867; Grassmann, 1853) in which the 3-dimensional nature of human color vision has its basis in the three different cone types in the human retina. Jacobs's recent review (Jacobs, 2018) of color dimensionality in animal vision, however, reminds the community that the dimensionality of color vision does not automatically equal the number of cone types, and that the most rigorous way to measure it remains Maxwellian color matching – as we follow in this work.

## 2.6 Complexity of Human Color Vision

This paper focuses on color dimensionality because it is the foundational characteristic of an observer's color experience, but many layers of further perceptual complexity exist atop that foundation. Examples include chromatic adaptation (color constancy) (Von Kries, 1902; Land, 1977; Foster, 2011), perceptual nonuniformity across colorspace (CIE, 1976; MacAdam, 1942), complex contextual interactions (Fairchild, 2013), and even surprising flood-fill features (Pinna, 1987; Pinna et al., 2001). Parts of these perceptual phenomena are scientifically mapped to neural correlates, such as nonlinear photoreceptor responses (Krauskopf & Karl, 1992; Angueyra et al., 2022), or parvocellular pathway and neural processes spanning V1, V2, and V4 (Livingstone & Hubel, 1987; Zeki, 1978; Li et al., 2014; Liu et al., 2020; Angueyra et al., 2022). This paper leaves these additional layers of perceptual complexity as future modeling work.

## 3 Simulation Engine for Biological Eyes and Optic Nerve Signals

We model the primary functions of the human retina based on the known science. The inputs to our retina model are hyperspectral images (Fig.2.A; details in Appendix A.1). A model of fixational eye drift (Rucci & Victor, 2015; Young & Smithson, 2021; Martinez-Conde et al., 2013) generates a sequence of frames that sample different parts of the image. Each frame is projected on the retina, stimulating a randomized array of cone cells, according to known cell density variation as a function of eccentricity (distance from the fovea) (Curcio et al., 1990) and known statistics of different cone cell types (Carroll et al., 2000) (Fig. 2.B). Each cone cell converts the scene light into photoreceptor activations based on cone type spectral response functions (Stockman et al., 1999; Stockman & Sharpe, 2000) (Fig. 2.C), followed by lateral inhibition from horizontal cells (Wool et al., 2018; Rodieck, 1965) (Fig. 2.D). These signals are transformed into spike trains (Fig. 2.E), then bundled into optic nerve signals (ONS), resulting in spatial distortion (Fig. 2.F), turning a color scene stimulus into a noisy, spatially distorted ONS. Additional details are in Appendix A and Video 1:35.

One observation to make is that there are three invariant properties of the simulated retina (i.e. retinal invariants): cell positions, cell types, and horizontal cell connections. These properties are held constant during generation of ONS for a particular eye, but we use the engine with different values for these properties to generate ONS datasets for a diverse set of eyes. For example, we create datasets with different cone types, including monochromatic (L, M, S), dichromatic (LM, LS,

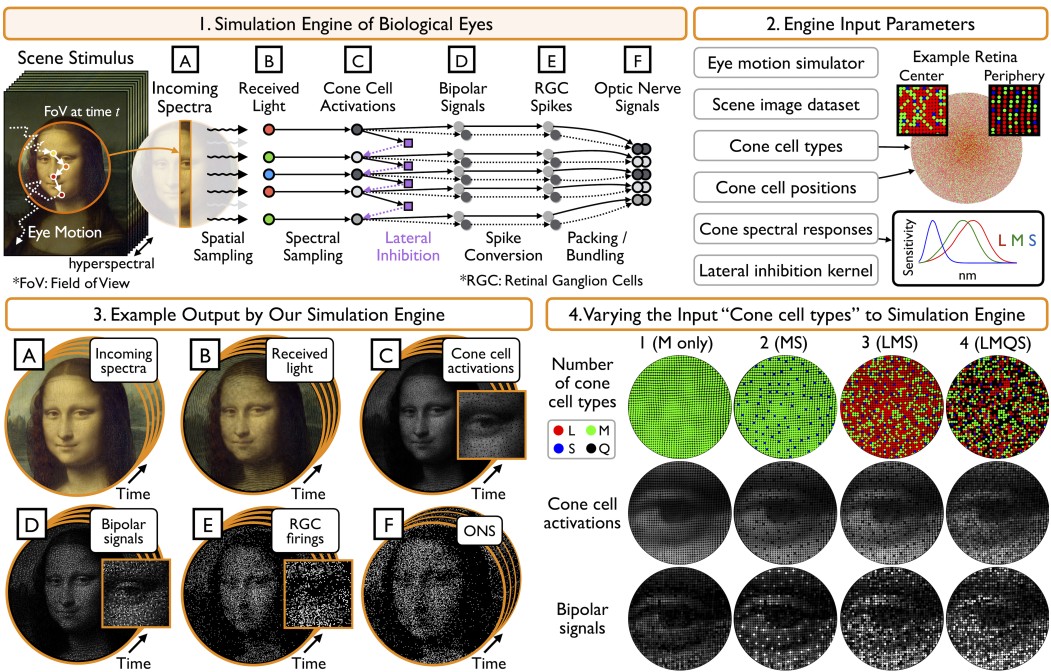

Figure 2: Overview of our simulation engine of biological eyes. 1. This engine takes a scene stimulus as an input, processes it through a "textbook" model of eye motion and retinal neural circuitry, to generate a stream of optic nerve signals. 2. This engine accepts custom eye and retina parameters. 3. It allows visualization of neural signals in steps (A-F), illustrating the progressive entanglement of scene imagery with retinal properties. 4. Visualization of changing one of the input parameters, the number of cone cell types – showing that the signals become noisier as the number increases.

MS), trichromatic (LMS) and tetrachromatic (LMSQ) configurations (Fig. 2.4). The generated ONS becomes spatially noisier as the number of cone types increases, and we study how the cortex infers the inherent color dimensionality of these eyes purely from the differences in their respective ONS.

## 4 SIMULATING CORTICAL LEARNING AND EMERGENCE OF COLOR VISION

Our cortical model is structured to learn three functions in a pipeline (Fig. 3.1): 1. $\Phi$ that decodes the optic nerve signal at time $t$ into its internal percept; 2. $\Omega$ that translates this percept according to eye motions inferred from the signal over a short time interval, $dt$; and 3. $\Psi$ that re-encodes the translated percept back into a predicted optic nerve signal at time $t + dt$, which is compared against the real signal received at that time. Mathematically, given the optic nerve signal $O_t$ at time $t$;

$$\hat{O}_{t+dt} = \Psi(\Omega(\Phi(O_t))) \tag{1}$$

where $\hat{O}_{t+dt}$ is the predicted ONS at time $t + dt$. Here the task of decoder $\Phi$ is analogous to inverting the retinal processes to transform $O_t$ into the visual percept image $V_t$ (i.e. $V_t \leftarrow \Phi(O_t)$). Likewise, the re-encoder function $\Psi$ resembles the retinal processes, as it aims to reproduce an optic nerve signal from the visual percept (i.e. $O_t \leftarrow \Psi(V_t)$). Therefore, $\Phi$ and $\Psi$ are pseudo-inverses.

The learning objective is to minimize the prediction error $E_{\text{prediction}}$, the difference between predicted and real optic nerve images at time $t + dt$, such that:

$$E_{\text{prediction}} = \|O_{t+dt} - \hat{O}_{t+dt}\|_2^2 = \|O_{t+dt} - \Psi(\Omega(\Phi(O_t)))\|_2^2 \tag{2}$$

where $O_{t+dt}$ is the real observed ONS at time $t + dt$.

Three main ideas drove the evolution of our selected model features. First, we reasoned that it would be a big step towards successful decoding and re-encoding if the cortex could infer the key encoding properties of each cone cell. We gave the cortical model sets of learnable parameters, which we

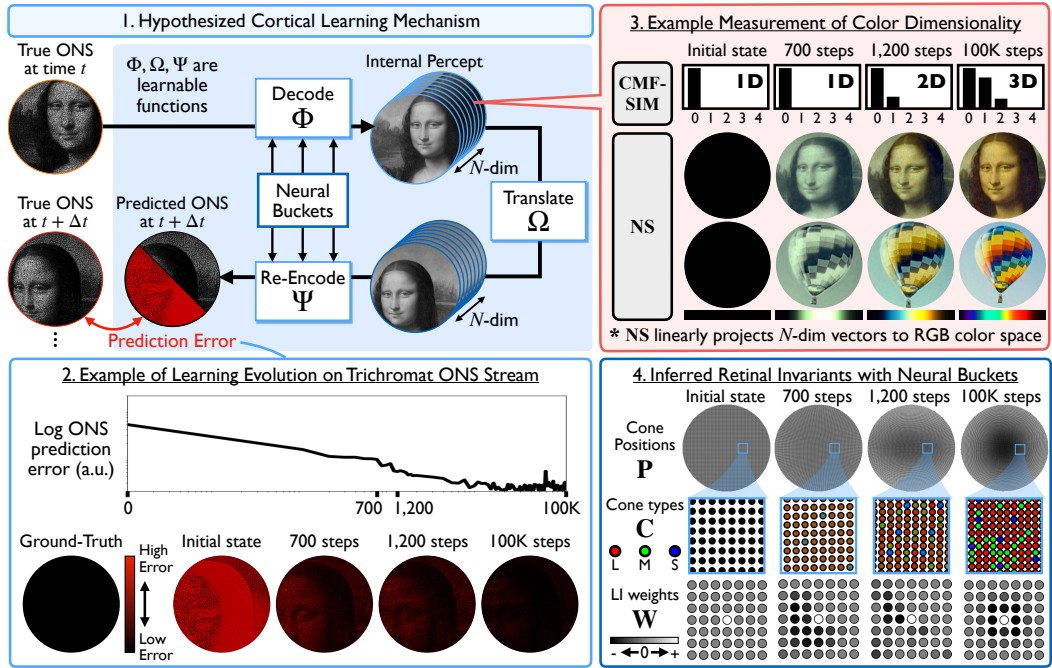

Figure 3: Overview of our hypothesized cortical learning mechanism and exclusive study of the learning behavior of the cortical model with trichromatic retina. 1. Given the stream of optic nerve signals as the only input data, the cortical model aims to predict the next ONS from the current one with 3 learnable functions, decoder $\Phi$, translation operator $\Omega$ and re-encoder $\Psi$. 2. Prediction error decreases as learning progresses, converging after 100K learning steps. 3. During learning, the color dimensionality of the internal percepts transition from 1D, 2D to 3D, formally measured by $CMF\text{-}SIM$ and visualized by $NS$ (Fig. 4 & Section 5). 4. The cortical model infers the retinal invariant properties during learning: cell positions P (higher density in fovea), cone cell types C, and lateral inhibition weights W (center-surround receptive field).

call "neural buckets", in which to store and update guesses of these properties during learning. The buckets contain the following information for each cone cell: 2D position in visual space (P), cone spectral identity (C), and lateral inhibition weights (W) to neighboring cones.

The second main idea was the observation that decoder $\Phi$ and encoder $\Psi$ are mathematically factorizable into a pipeline of subfunctions, such that:

$$\Phi = \Phi_P \circ \Phi_C \circ \Phi_W$$
$$\Psi = \Psi_W \circ \Psi_C \circ \Psi_P.$$

Each sub-function is an operator conditioned on its corresponding neural bucket, cell positions P, cone spectral types C, and lateral inhibition weights to neighboring cells W. In case of decoder $\Phi$, it first executes inversion of lateral inhibition using $\Phi_W$ in order to estimate the activations of each photoreceptor associated with an optic nerve axon; second, projects scalar cone activations into $\mathbb{R}^N$ by $\Phi_C$ using inferred spectral identities in C (and interpolating color across space – the third main idea below); and finally inverting the spatial distortion of foveation via $\Phi_P$. The re-encoding function is a pipeline of analogous subfunctions in reverse order: re-applying spatial warping with $\Psi_P$; re-projecting color into scalar photoreceptor activations with $\Psi_C$; and re-applying lateral inhibition with $\Psi_W$. Further implementation details are provided in Appendix B.

The third main idea was that, in order to accurately re-encode after translation, the model needs to learn to interpolate color information spatially, because there is only one cone type at each point on the retina. This need to interpolate is analogous to the situation in cameras with image sensors (Bayer, 1976; Fossum, 1997; Kimmel, 1999) that physically sample only one of the R,G,B channels at each pixel, and fundamentally require demosaicking algorithms to interpolate full R,G,B values at all pixels. The required interpolation function in the cortical model is more complex because the spectral

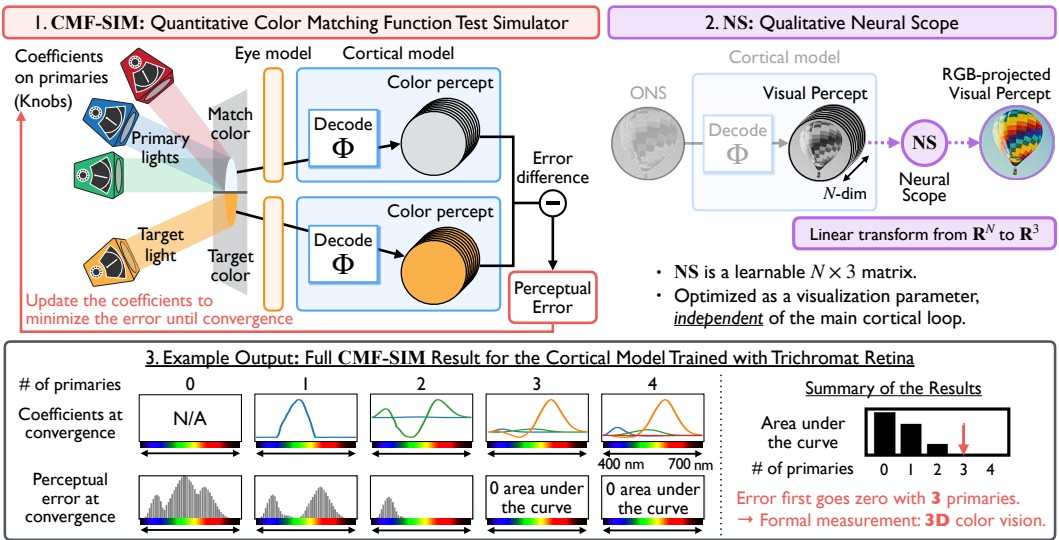

Figure 4: Overview of our measurement methods of emergent color dimensionality, Color Matching Function Test Simulator (*CMF-SIM*) & Neural Scope (*NS*). 1. *CMF-SIM* determines the minimum number of primary colors needed to match any target color by iteratively updating coefficients to minimize perceptual error. 2. *NS* visualizes visual percepts independently of the cortical learning loop, optimized as a learnable $N \times 3$ matrix to minimize projection error to the target RGB image. 3. Example *CMF-SIM* output for a trichromat retina-trained cortical model shows matching errors for 400–700nm spectral light converging to zero with three primaries, confirming 3D color vision.

sampling pattern is random, so learning this function is entangled with correctly resolving C. We enable and force the cortical model to learn the color interpolation function by representing it as a convolutional neural network with neural bucket parameters D (Appendix B.2.2).

An important detail is the handling of eye motion in the simulation. In one experiment we show that the model can learn a subfunction that estimates the eye motion translation between times $t$ and $t + dt$ purely from the optic nerve signals at those times. The main computational challenge here is the spatial warp in optic nerve signals. The uniform translation in stimulus space corresponds to a non-uniform translation in the optic nerve signal space, which makes the prediction of the eye motion dependent on the inference of cell positions P. We find that the cortical model iteratively updates the neural bucket P from imperfect initial eye motion estimate, which helps to improve the prediction of eye motion, and vice versa. This inference converges to the correct eye motion estimate, as the model learns to minimize prediction error (Appendix B.3).

With this pipeline of subfunctions and associated neural buckets, the hypothesized learning is equivalent to parallel numerical optimization of all neural buckets in striving to minimize prediction error. We simulate learning using stochastic gradient descent (Kingma & Ba, 2014).

## 5 NEURAL REPRESENTATION OF COLOR SPACE AND ANALYSIS OF EMERGENT COLOR DIMENSIONALITY

We model color in the cortex as a vector in high-dimensional space, $\mathbb{R}^N$. This decision represents our view that the brain has both the freedom and the challenge of somehow inferring the intrinsic color dimensionality of the visual signals it is receiving along the optic nerve. Specifically, we define a cortical decoding function $\Phi$ that takes the optic nerve signal $O_t$ at time $t$ and transforms it into its internal visual percept $V_t$ (Fig. 3), such that:

$$V_t = \Phi(O_t) \tag{3}$$

where each pixel in $V_t$ is a $N$-dimensional vector. Inside $\mathbb{R}^N$, we assume that cortical color space emerges as a $K$-dimensional manifold. To measure this intrinsic dimensionality $K$, we introduce two methods: formal, numerical *CMF-SIM*; and intuitive, visual *NS*.

*CMF-SIM* (Color Matching Function Test Simulator) is our tool to formally quantify the color dimensionality that has emerged in the $\mathbb{R}^N$ color space of the cortical model (Fig. 4.1). *CMF-SIM* treats the retina model and cortical model as a black-box color observer. Specifically, we limit ourselves to showing the model two patches of scene color at random, distinct locations on the retina to reflect the effect of real-world viewing conditions, obtaining only a scalar score as feedback to indicate the difference in color appearance betweem the two patches (with zero representing a color match). This interface is intentionally limited and identical to color matching experiments with human subjects. Then, resting on the formal, technical bedrock for colorimetry established by Maxwell (1856) and Grassmann (1853), we exhaustively probe to determine the minimum number of color primaries needed to match any test color through linear combination (Fig. 4.1 & Appendix C). For example, Maxwell found that most humans require 3 primaries and are formally trichromatic, but red-green colorblind persons require only 2 primaries and are dichromatic. We model diverse color observers and measure the color dimensionality of their emergent vision.

*NS* (Neural-Scope) is our tool to display emergent colors in $\mathbb{R}^N$ by projecting them into RGB color space (Fig. 4.2). *NS* is a learnable $N \times 3$ matrix mapping the emergent cortical color manifold in $\mathbb{R}^N$ linearly to conventional RGB color, enabling visual inspection of emergent color vision. To compute *NS*, we first project our hyperspectral image dataset in two ways: (1) using the retina and learned cortical model to map hyperspectral images into the $\mathbb{R}^N$ cortical colorspace, and (2) using conventional color processing to convert hyperspectral data to RGB via CIEXYZ (CIE, 1931). *NS* is then determined as the least squares transform from the former data in $\mathbb{R}^N$ to the latter in RGB. Importantly, *NS* is independently optimized as a visualization parameter, separate from the main cortical loop, ensuring that the cortex processes ONS without any exposure to input images in either hyperspectral or RGB form. *NS* provides striking color visualizations and complementary intuition visually, which are fully consistent with formal *CMF-SIM* results. For example, *NS* allows us to compute and contrast the color palettes of the three different types of human dichromacy (Fig. 5).

In sum, we model cortical color as vectors in $\mathbb{R}^N$, allow color space to emerge naturally through the hypothesized learning mechanism, and measure the emergent color space's dimensionality quantitatively with *CMF-SIM* and analyze it visually with *NS*.

## 6 SIMULATION RESULTS - EMERGENCE OF COLOR VISION

Figure 3.2 begins with a model of typical human retina containing L, M and S cones, and illustrates the time-varying behavior of the cortical model as it learns color vision. The visualized prediction error decreases as the training progresses, and the internal percept converges to 3-D color vision both formally (*CMF-SIM*) and visually (*NS* imagery), shown in Figure 3.3. Notably, the visual timeline highlights that the cortical model learns color vision one dimension at a time: achieving monochromacy at 700 learning steps, dichromacy at 1,200 steps, and converging to trichromacy after $10^6$ steps. At convergence, the cortical model has accurately inferred all retinal properties: spectral identity of each cell, cell positions and lateral inhibition neighbor weights (Fig. 3.4). For a cortical model trained with a trichromat retina, *CMF-SIM* results closely align with human psychophysical data (Stiles & Burch, 1955) (Appendix G.1) and the model consistently demonstrates 3D color vision across different noise initializations (Appendix G.2), highlighting our model's validity and robustness.

The remainder of this section shows *CMF-SIM* and *NS* results at convergence for a diversity of simulation scenarios. Fig. 4.2 dissects the *CMF-SIM* analysis for the case where the retina contains 3 cone types. The graphs show color matching function results using optimal sets of color primaries from 0 to 4 primaries, along with the residual perceptual error as a function of wavelength. The area under the curve (AUC) for each error graph suggests that the error falls to near-zero only with at least 3 primaries – this formally proves that the color dimensionality is 3.

Figure 5 presents results of the hypothesized cortical learning in a diversity of color observers where the retina contains different numbers of cone cell types. This shows that the model learns $K$-dimensional color vision when the retina contains $K$ cone types. That is, when the retina contains 1, 2, 3 or 4 cell types, the cortical model converges on mono, di, tri, or tetrachromat color vision, formally quantified with *CMF-SIM* (further analysis of tetrachromat models in Appendix F). And qualitatively, we observe that the *NS* images are grayscale for $K = 1$, colorblind with only shades of blue and yellow for $K = 2$, and full color with all trichromatic hues only with $K = 3$ (Fig. 5.1). Use of *NS* is limited to color dimensionality up to 3, and is to not applicable to $K = 4$ case, but

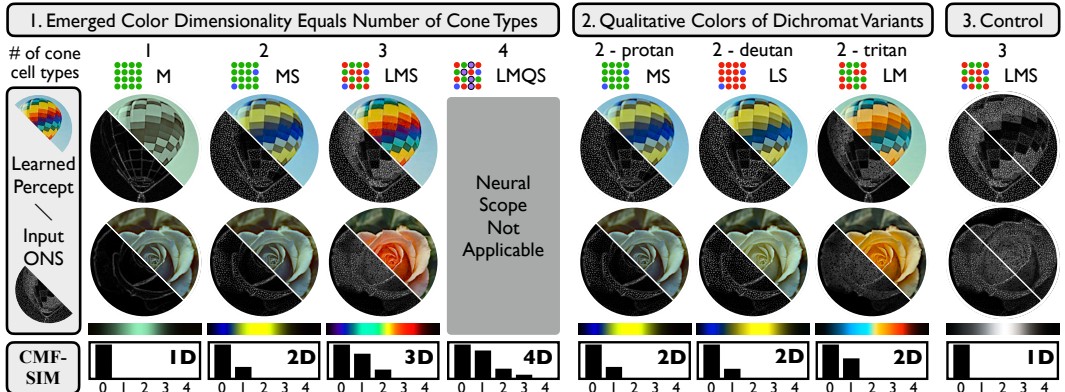

Figure 5: Results of simulating emergence of color vision from various retinas, with analysis of learned color dimensionality using qualitative visualization ($NS$) and formal methods ($CMF$-$SIM$). 1. Cortical models trained with dataset generated with retinas containing 1, 2, 3, 4 cone types result in mono-, di-, tri, tetrachromatic color vision, respectively. 2. Qualitative color of dichromat variants is consistent with known vision science on color vision deficiency. 3. Control experiment with a trichromat retina, but with cortical learning deliberately removed: $CMF$-$SIM$ measures color as 1-D, highlighting that cortical learning is necessary for emergence of correct color dimensionality.

$CMF$-$SIM$ formally confirms that 4-dimensional color emerges there. All variants of retinas with two cone types (protanopia, deuteranopia and tritanopia), converge to 2D color vision. But more striking, $NS$ reveals hue shifts among these models that are consistent with color vision deficiencies studies (Brettel et al., 1997) (yellow-blue hues for deuteranopia / protanopia (Judd, 1948; Graham & Hsia, 1959) and teal-pink hues for tritanopia (Alpern et al., 1983)) (Fig. 5.2).

In a control experiment, we verify that cortical learning is essential for color vision by comparing two scenarios: (1) a baseline where optic nerve signals directly form percepts, with all cortical learning deliberately removed (Fig. 5.3), and (2) the proposed model including cortical learning (Fig. 5.1, LMS case). The control baseline experiment fails standard color-matching tests, reducing color dimensionality to 1-D despite 3 cone types, demonstrating that cortical learning is indeed necessary for emergence of correct color dimensionality (details in Appendix D). In contrast, the proposed learning-based model results in correct 3-D color, as expected.

Figure 6 simulates boosting of color dimensionality in adulthood by gene therapy. Previous genetic studies (Jacobs et al., 2007; Mancuso et al., 2009; Zhang et al., 2017) demonstrated that the introduction of an additional class of photopigments in the mammalian cone mosaic, even in adulthood via gene therapy, resulted in a new dimension of chromatic sensory experience. Here, we simulate the experiment performed in squirrel monkeys (Mancuso et al., 2009), finding simulation results consistent with the noted boost in color dimensionality (Fig. 6.1). First, we model the vision of an adult male squirrel monkey with a protanopic retina (M and S cones only) and a cortical model that has converged. Next, we simulate the effects of gene therapy, by modifying the retina model so that a random subset of cones begin to express L opsin. Immediately after this retinal change, $NS$ continues to show blue-yellow dichromatic vision. However, if we allow the cortical model to continue the hypothesized self-supervised learning, vision re-converges to boosted, 3-dimensional color (Fig. 6.1). As an aside, the real gene therapy experiment (Mancuso et al., 2009) resulted in expression of both M and L photopigment in affected cones, but the relative amounts are difficult to ascertain. We additionally model scenarios in which equal M and L expression occurs or a variable amount of L relative to M at each cell, and both scenarios converge to trichromacy as well (Appendix E). These result indicate that the hypothesized self-supervised learning can explain experimental boosting of color dimensionality in adulthood consistent with Mancuso et al. (2009), if the hypothesized learning is assumed to occur continuously even in adulthood.

Figure 6.2 further demonstrates that simulations of boosting color dimensionality succeed even in 3D→4D cases by the addition of a fourth cone between S and M cones. This simulation results present the first theoretical work that highlights the possibility of boosting humans with trichromatic

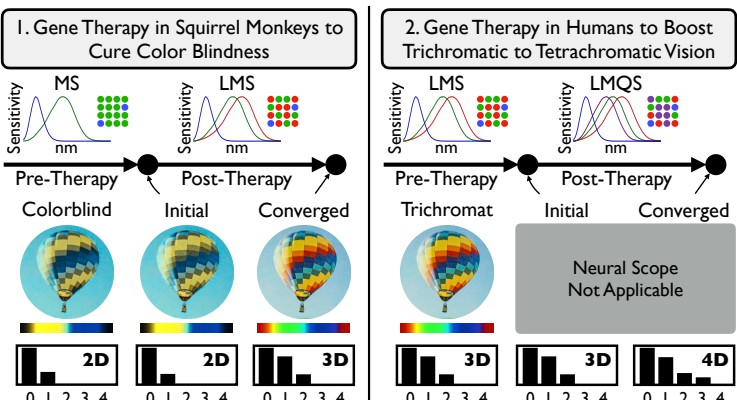

Figure 6: Simulated experiments for boosting color dimensionality via gene therapy. 1. 2D dichromat model is boosted to 3D trichromacy by mutating some M cones to express L opsins, with cortical learning re-converging to 3D color. 2. 3D trichromat model is boosted to 4D tetrachromacy by adding a fourth cone type between M and L cones, with cortical learning re-converging to 4D color.

vision to tetrachromat one by addition of a fourth photopigment. Further details, including the study of human tetrachromat observer model, are described in Appendix F.

# 7    CONCLUSION

In this work, we presented a framework for modeling the emergence of color vision in the human brain. We introduced a simulation engine for optic nerve signals, simulated cortical learning purely from such input signals, and measured the emergent color dimensionality qualitatively with $NS$ and quantitatively with $CMF\text{-}SIM$.

We believe that the critical contribution of this work actually lies in the computational formulation of the problem in human perception itself – given access to only a stream of optic nerve signals, we probe how to meaningfully model the emergence of color vision in the cortex via simulated learning. Once the problem is formulated this way, it may not be entirely surprising to machine learning researchers that color with the correct dimensionality can be inferred; however, from the vision science perspective, this new approach is in many ways a foreign way to formulate the problem, because it is more common to think color comes from hardwired neural circuits (Appendix D).

The connection to camera imaging systems is noteworthy as well. The proposed computational framework is akin to a camera processing pipeline attached to an unknown, random color filter array pattern, where even the number of different color filters is a mystery. This is related to a branch of computational imaging called "auto calibration" that jointly solves for the scene and unknown system calibration parameters from measurements. One could imagine a new class of engineered sensing systems that have general processing units along the lines of the learning mechanism in this paper, that enables perceptual inference from a broader range of sensory streams that do not require the precision manufacturing common to many current sensors and cameras.

The learning notion of perception emerging from a process of disentangling it from encoded sensory streams is an interesting intellectual view of how perception may generally emerge in the brain. Any sensory stream going to the brain encodes information from the world entangled with sensing organ characteristics. The self-supervised learning mechanism proposed in this paper might be abstracted into a general neural process of learning to predict fluctuations in sensory stream due to ego perturbation, with perception for that sense emerging neurally as the optimal internal representation and associated decoder/encoder pair that enables accurate prediction.

To conclude, we invite the research community to build on the computational framework proposed in this paper, by improving the visual system components (e.g. even more accurately modeling details in the eye model, or replacing back-propagation in the cortical model with more bio-plausible learning rules (Lillicrap et al., 2020; Hinton, 2022)), or applying the framework to other sensory modalities.

ACKNOWLEDGEMENTS

This project was supported by Air Force Office of Scientific Research's MURI grant (FA9550-20-1-0195 and FA9550-21-1-0230). We are grateful to everyone involved in the MURI project for their support and feedback. Special thanks to Bruno Olshausen, Jessica Lee, and Alexander Belsten for their valuable discussions and insights. We also appreciate Fred Rieke, Lawrence Sincich, and David Brainard for their thoughtful input and engaging conversations.

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

# Supplementary Materials

In this supplementary section, we provide additional details on our computational framework for modeling the emergence of human color vision. Section A covers the simulation of optic nerve signals from hyperspectral scene images and our modeling of retina circuitry. Section B details the design of our simulated cortical model. Section C provides a step-by-step description of how we simulate formal color matching function tests of color dimensionality, $CMF\text{-}SIM$. In Section D, we probe whether the emergence of color vision depends on cortical learning, by introducing comparative, baseline cortical models. Section E simulates variant cell expression phenotypes after gene therapy in squirrel monkeys (Mancuso et al., 2009), showing robustness of the simulation result that addition of a new cell opsin gene enhances color dimensionality. Finally, Section F discusses our tetrachromacy simulation, highlighting the differences between the human Q cone (between M and L cones) and the pigeon Q cone (between S and M cones). A supplementary video is available on our project website: https://matisse.eecs.berkeley.edu.

Table 1: Glossary of terms used in our framework

| | |
|---|---|
| **Scalars** | |
| $t, x, y$ | Time, world view coordinates |
| $u, v$ | Optic nerve image coordinates |
| $N$ | $N$-dimensional vector represents cortical color |
| **Eye Simulation Engine** | |
| $S_t(x, y)$ | Scene image, spectrum at each pixel |
| $\Upsilon$ | Retina encoding of scene into time-averaged optic nerve image: $\Upsilon(S_t) = O_t$ |
| **Cortical Image Functions** | |
| $O_t(u, v)$ | Time-averaged optic nerve image |
| $A_t(u, v)$ | Cone activations image |
| $\tilde{V}_t(u, v)$ | Visual percept image |
| $V_t(x, y)$ | Unwarped visual percept image |
| $\hat{O}_{t+dt}(u, v)$ | Predicted optic nerve image, time $t + dt$ |
| **Neural Buckets (Learnable Parameters)** | |
| C | Color type of each cone |
| P | Position of each cone in visual field |
| W | Lateral inhibition weights to neighboring cones |
| D | Demosaicing operator parameters |
| M | Motion estimation operator parameters |
| **Learned Cortical Function Operators** | |
| $\Phi, \Psi$ | $\Phi$ decodes $O_t$ into $V_t$, and $\Psi$ re-encodes $V_t$ into $O_t$. $\Phi = \Phi_P \circ \Phi_C \circ \Phi_W$, and $\Psi = \Psi_W \circ \Psi_C \circ \Psi_P$. |
| $\Phi_W, \Psi_W$ | Transforms $O_t$ into $A_t$, and vice-versa. |
| $\Phi_C, \Psi_C$ | Transforms $A_t$ into $\tilde{V}_t$, and vice versa. |
| $\Phi_P, \Psi_P$ | Transforms $\tilde{V}_t$ into $V_t$, and vice versa. |
| $\mu_M(O_1, O_2)$ | Estimates world translation $(x, y)$ between two optic nerve images |
| $\Omega_{(x,y)}$ | Translates an image laterally by $(x, y)$. $\Omega_{\mu_M(O_t, O_{t+dt})}(V_t) = V_{t+dt}$ |

## A  SIMULATION ENGINE OF BIOLOGICAL EYES

We simulate the optic nerve signals from hyperspectral scene images, simulate eye gaze movements, and model the retina circuitry. We describe the details of these simulations in the following.

### A.1  DATABASE OF HYPERSPECTRAL SCENE IMAGES

We model the scene viewed by the eye using a dataset comprising 900 hyperspectral images from everyday scenes, captured with a hyperspectral camera (Arad et al., 2022). Each image in the dataset

is of dimension 512 pixels × 482 pixels × 31 spectral channels, where each pixel is represented by its spectral power distribution ("spectra " below). To generate our training data set, we cropped thousands of 482x482 patches from these hyperspectral images, ensuring sufficient resolution to avoid aliasing effects, induced by varying cone cell densities (Section A.3.2).

While only hyperspectral images are used during learning simulation, RGB images can be presented during test-time by simulating a projector with specific spectral power distributions (SPDs) for the R, G, and B channels (Cottaris et al., 2019), effectively converting each RGB image into a spectral representation before passing into the retina model. This test-time use of RGB images does not affect learning and is purely for illustrative purposes in the figures (e.g., Figure 3).

## A.2 Eye Gaze Movements – Fixational Drift

We simulate small eye movement as fixational eye drift with a random walk (Young & Smithson, 2021). This produces a stream of the current scene image translating across the retina. Over a time interval of duration $dt$, we sample the change in eye motion from a uniform probability distribution $(dX, dY) \sim \mathcal{U}\{(-15, 15) \times (-15, 15)\}$ where 15 corresponds to 15 pixels in the scene image $S$.

The Mona Lisa video in Supplementary Video 2:08 also incorporates manually-authored saccades (e.g. from hand to mouth) for illustrative purposes, but the learning simulation uses only fixational drift.

## A.3 Retina Circuitry Model

Our model of light detection and signal encoding in the retina is the textbook model (Rodieck, 1998) of the photopic, midget, "private-line" pathway (see Figure 2). Details of our implementation follow.

### A.3.1 Cone Mosaic and Spectral Sampling

In our simulation, we model $256 \times 256$ cone cells, with the random distribution of L, M and S cone cells on the retina with a relative probability of 0.63, 0.32 and 0.05 from Sabesan et al. (2015). We model L, M and S spectral response using the template response function from Carroll et al. (2000), with spectral peaks of 560, 530, 419 nm, respectively. We model photon-shot noise in the photoreceptor activation values, simulating a signal-to-noise ratio of approximately 100. The simulation omits foveal tritanopia (Williams et al., 1981).

### A.3.2 Foveation and Cone Positions

We model cone cell spatial density on the retina in accordance with Curcio et al. (1990), while enforcing retinotopy. Starting from a regular grid of cells, we computationally perturb positions until the target spatial distribution is stochastically achieved, while constraining each cell to be always confined by its neighbors. We model biological variation in cell locations, which represents a challenge for cortical learning, by adding multi-resolution positional randomness (Perlin, 1985).

### A.3.3 Center-Surround Lateral Inhibition

We mathematically model textbook lateral inhibition in retinal signals as a Difference-of-Gaussians (DoG) convolution kernel (Enroth-Cugell et al., 1983), with parameters fitted from electrophysiology (Wool et al., 2018). Specifically our DoG kernel has standard deviations of 0.15 and 0.9 cone diameters, respectively, for the positive center and negative surround Gaussians. Significantly, the relative amplitudes are such that the kernel has a non-zero mean of 0.09 (Lennie et al., 1991; Wool et al., 2018). We convolve the array of cone activation values by this DoG kernel to compute outputs to bipolar cells.

### A.3.4 On/Off Pathways and Optic Nerve Spiking

We model textbook on- and off- connections from cone activations to bipolar cells, forwarding to on- and off- retinal ganglion cells (RGC)s. The on- activation is modeled by a rectified linear unit activation function, $\text{ReLU}(x) = \max(0, x)$, where $x$ is the laterally-inhibited cone output. The off-activation is modeled as $\text{ReLU}(-x)$. Optic nerve spikes are the outputs at RGC axons, with action

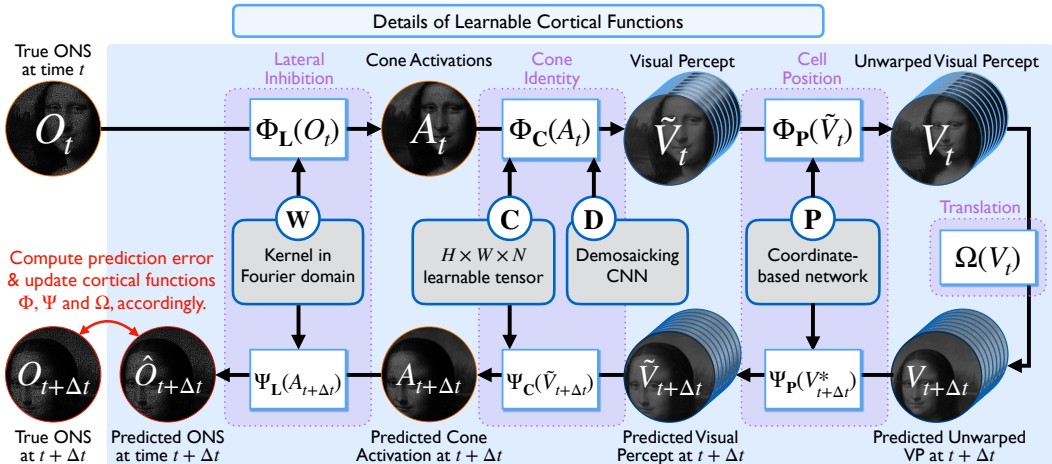

Figure 7: Pipeline of decoding, translation and re-encoding cortical functions. The optic nerve signal image $O_t$ is decoded into visual percept image $V_t$. The percept is translated in accordance with eye motion over time $dt$, then re-encoded back into a prediction of the optic nerve signal $\hat{O}_{t+dt}$ at a short time in the future.

potentials are generated with Leaky Integrate and Fire model (Lapicque, 1907; Abbott, 1999; Arbib, 2003).

## B    CORTICAL MODEL

### B.1    TIME-AVERAGED OPTIC NERVE SIGNALS

We model the first step of cortical processing as re-combination of the on- and off- RGC action potentials followed by time-averaging. This results in an image stream $O_t$ with real-valued pixel values that are proportional to the laterally-inhibited output from cone cells. $O_t$ is the sole input to the remainder of the cortical model.

### B.2    PREDICTION PIPELINE AND LEARNING OBJECTIVE

Figure 7 shows an elaboration of the cortical prediction pipeline introduced in Figure 3.1 and summarized in Section 4. This pipeline represents an existence-proof simulation that the hypothesized self-supervised learning on $O_t$ can successfully produce: (1) emergence of vision with the correct color dimensionality, and (2) inference of invariant retinal properties, including learned neural buckets for cone spectral identities C, cell positions P and lateral inhibition weights W. Our results show that both can be simultaneously achieved by learning to minimize the error in predicting the fluctuations in optic nerve signal values.

To re-summarize Section 4, the prediction is made by applying the learned functions for decoding $\Phi$, translation $\Omega$ and re-encoding $\Psi$ to the optic nerve signal $O_t$, to obtain a predicted optic nerve signal $\hat{O}_{t+dt} = \Psi(\Omega(\Phi(O_t)))$. The learning objective function is to minimize prediction error, the difference between predicted and real optic nerve images at time $t + dt$: $E_{\text{prediction}} = \|O_{t+dt} - \hat{O}_{t+dt}\|_2^2 = \|O_{t+dt} - \Psi(\Omega(\Phi(O_t)))\|_2^2$. The decoder $\Phi$ and re-encoder $\Psi$ functions are each factorized into a pipeline, such that: $\Phi = \Phi_P \circ \Phi_C \circ \Phi_W$ and $\Psi = \Psi_W \circ \Psi_C \circ \Psi_P$, where each sub-function is an operator conditioned on its corresponding neural bucket, C, P and W. The following sections describe the implementation of each learnable neural bucket and its associated cortical sub-function.

### B.2.1    LEARNING LATERAL INHIBITION NEIGHBOR WEIGHTS W

In the factorized function pipeline, sub-decoder $\Phi_W$ conceptually inverts lateral inhibition to transform optic nerve image $O_t$ into cone activation image $A_t$. Conversely, sub-encoder $\Psi_W$ reproduces lateral

inhibition to transform $A_t$ into $O_t$. These operators are modeled, respectively, as inverse and forward convolution operators, implemented by pixel-wise division and multiplication in the Fourier domain. The learnable parameters for this operator are neural bucket W, representing the Fourier transform image pixels of the lateral inhibition DoG kernel's Fourier transform. The resolution of W is set equal to the image resolution of $O_t$.

### B.2.2  LEARNING COLOR TYPES C AND INTERPOLATION D

In the factorized function pipeline, sub-decoder $\Phi_C$ conceptually transforms cone activation image $A_t$ (scalar-valued pixels) into a full-color visual percept image $\tilde{V}_t$ ($N$-dimensional vector pixels). Conversely, $\Psi_C$ re-encodes $\tilde{V}_t$ into $A_t$. These functions depend on two neural buckets of learnable parameters, C and D. C is an image of $N$-dimensional vectors, in which the cortical model learns to represent the spectral type of the source cone associated with each pixel in $A_t$. D are the parameter weights of a learnable function $\phi_D$ that spatially interpolates color across the image, which we implement as a convolutional neural network based on the U-Net (Ronneberger et al., 2015) architecture.

Mathematically, $\Phi_C$ and $\Psi_C$ are defined as $\Phi_C(A_t) = \phi_D(A_t \otimes C)$, where $\otimes$ denotes element-wise multiplication; $\Psi_C(\tilde{V}_t) = \tilde{V}_t \otimes C$, where $\odot$ denotes element-wise dot product. To maintain $\Phi_C$ and $\Psi_C$ as pseudo-inverses, we constrain $C \odot C$ to be an image with all pixels equal to 1.

### B.2.3  LEARNING CELL POSITIONS P

As shown in Fig. 2.3.F, optic nerve images are spatially distorted due to foveation, whereas visual perception is apparently undistorted. In the factorized function pipeline, sub-decoder $\Phi_P$ conceptually transforms $\tilde{V}_t(u, v)$, which is foveated and spatially distorted, into the final visual percept image $V_t(x, y)$, which is Euclidean and undistorted. Conversely, $\Psi_P$ re-warps $V_t(x, y)$ into $\tilde{V}_t(u, v)$.

We define a learnable spatial warping function $\phi_P(u, v) = (x, y)$, implemented using normalizing flow (Rezende & Mohamed, 2015; Dinh et al., 2016) that is an invertible function. Neural bucket P comprises the learnable parameters of this normalizing flow network. Then $\Phi_P(\tilde{V}_t)(x, y) = \tilde{V}_t(\phi_P^{-1}(x, y))$, and $\Psi_P(V_t)(u, v) = \tilde{V}_t(\phi_P(u, v))$.

### B.3  LEARNING EYE MOTION ESTIMATION M

In the learning loop, cortical sub-function $\Omega$ translates the visual percept image according to the spatial motion $(dx, dy)$ that occurs during the brief period between $t$ and $t + dt$. Where does $(dx, dy)$ come from? Here, we describe how the cortex can learn a helper cortical function $\mu_M(O_t, O_{t+dt})$ that estimates $(dx, dy)$ directly from the optic nerve stream values at times $t$ and $t + dt$, maintaining the strict operation of the cortical learning model purely from optic nerve signals.

We model learning of $\mu_M(O_t, O_{t+dt})$ with neural bucket M by optimizing:

$$M = \underset{M,P}{\operatorname{argmin}} \|B(O_{t+dt}) - \Psi_P(\Omega_{\mu_M(O_t, O_{t+dt})}(\Phi_P(B(O_t))))\|_2^2,$$

where B is a convolution operator that low-pass filters the optic nerve images, effectively halving the highest frequencies; $\Phi_P$ aims to dewarp the optic nerve image into Euclidean coordinates, and $\Psi_P$ is the inverse operator that rewarps the image. Details are discussed also in Supplementary Video 16:25.

### B.4  EFFICIENT LEARNING IMPLEMENTATION & ALTERNATIVE REPRESENTATION OF INTERNAL PERCEPTS

We simulate the self-supervised learning by parallel numerical optimization of all neural buckets by applying the stochastic gradient descent algorithm to minimize the prediction error metric. In practice, this is implemented by repeating the following step thousands of times until convergence: pick a batch of different scene images from the database; generate the optic nerve signals at time $t$ with the retina encoding model; send this into the cortical model and execute the decode / translate / re-encode functions with current neural bucket parameters to estimate the signal at $t + dt$; compute the actual optic nerve signal at $t + dt$ with the retina encoding model; compute the prediction error

by taking the difference between prediction and actual signal; backpropagate the prediction error to update neural bucket parameters.

For efficient processing of cortical functions, we avoid computing full-resolution undistorted visual percept images $V_t(x, y)$ during the learning simulation. Instead, we directly compute $\tilde{V}_{t+dt}(u, v)$ by optical flow of the pixels in $\tilde{V}_t(u, v)$, where the optical flow map is defined by $(\Psi_P \circ \Omega_{\mu_M(O_t, O_{t+dt})} \circ \Phi_P)$. This is mathematically equivalent to the learning implementation described in Section B.2, and the cortical model continues to learn cell positions P, and therefore decoder $\Phi$ can still be applied to compute undistorted visual percepts when desired.

## C    Color Matching Function Tests & *CMF-SIM*

In the classical color matching theory of Maxwell (1856) and Grassmann (1853), an observer compares a test color spectrum, $t$, and a set of $K$ primary color spectra, $p_i$ for $i = 1, ..., K$ and tunes real weights $\alpha_i$ for each primary until a color match is achieved. If a particular weight has a negative value, primary light of that magnitude is added to the test color $t$ rather than to the other primary colors. Mathematically, a color match is achieved when the matching spectrum $\sum_{i=1}^{K} \alpha_i^+ p_i$ and test spectrum $t + \sum_{i=1}^{K} (-\alpha_i)^+ p_i$ appear identical. Here, $\alpha_i^+$ is defined as $\max(\alpha_i, 0)$.

We simulate such classical color matching by formulating the retina model plus learned cortical model as a black-box color observer in the colorimetric sense. That is, we take the test and matching spectra, simulate the stimulation on different parts of the retina for test and match color patches, apply the retina encoding model $\Upsilon$ and the learned cortical decoder $\Phi$ to each separately, and iteratively update the weights $\alpha_i$ until there is no further improvement in the computed perceptual difference $E = \|\Phi(\Upsilon(\sum_{i=1}^{K} \alpha_i^+ p_i)) - \Phi(\Upsilon(t + \sum_{i=1}^{K} (-\alpha_i)^+ p_i))\|_2^2$. Note that the optimal $E$ will not be close to zero if the primaries cannot match test color $t$.

The color dimensionality of an observer is formally equal to the minimum number of primary colors required to match any test color. *CMF-SIM* simulates thousands of color matches to rigorously compute this dimensionality for any given retina model plus cortical model. In *CMF-SIM*, we simulate classical color matching function (CMF) tests of exhaustively attempting to match test colors equal to each monochromatic wavelength (100 samples from 400 to 700 nm), with a linear combination of $K$ primary colors. For the primary colors, we use distinct, monochromatic spectra. We start CMF tests with a single primary, and add primaries one-by-one until all test wavelengths in the CMF can be matched successfully. The pseudocode for *CMF-SIM* is described in Algorithm 1.

For example, for our cortical model to be formally measured as trichromat, we must show that there do not exist any 1 or 2 primaries that can pass the CMF tests, and show 3 primaries that succeed. A brute-force approach would be to exhaustively test all possible sets of $K$ primaries; for efficiency, we instead perform stratified sampling to randomly generate a large number (e.g. 500) distinct sets of primaries for each choice of $K$. We require that all of these possible primary sets fail, with the aggregate perceptual error across all wavelengths being greater than a threshold $\epsilon$, before incrementing the CMF tests to $N$+1 primaries.

## D    Baseline Results – Testing if Cortical Inference is Required for Color Vision

A fundamental question about color perception is whether cortical inference is necessary, or if human color perception arises directly from hardwired neural circuits. The main paper presents analysis an existence proof that cortical inference can indeed result in color vision of the correct dimensionality. Here, we add evidence that cortical inference of some kind is necessary, by modeling and testing three baseline cortical models with limited or no cortical processing.

For our first baseline, we assume an extreme case where no cortical learning occurs, treating raw optic nerve signals directly as internal percepts (i.e., the cortical decoder function $\Phi$ is an identity function). We apply *CMF-SIM* directly to the optic nerve signals generated by a trichromat retina. Figure 8.1 presents analysis that this model fails to generate vision that can be recognized as color consistent. The analysis is to perform baseline color matching experiments where the test and match

---

**Algorithm 1:** Pseudocode for *CMF-SIM*

---

**Data:** WAVELENGTHS, MAX_TRIALS, Retinal process $\Upsilon$, Cortical decoder $\Phi$

**Function** `CMF-SIM()`:
  | base_errors $\leftarrow$ `ComputeBaseErrors()`
  | color_dimensionality $\leftarrow$ `FindMinimumPrimaries(`*base_errors*`)`
  | **return** color_dimensionality

**Function** `ComputeBaseErrors()`:
  | Initialize an array errors
  | **foreach** *wavelength $\lambda$ in WAVELENGTHS* **do**
    | $(x, y)_A$ = RandomLocation()
    | $(x, y)_B$ = RandomLocation()
    | colorPatchA $\leftarrow$ MonochromaticColorPatch($\lambda$, $(x, y)_A$ )
    | colorPatchB $\leftarrow$ MonochromaticColorPatch($\lambda$, $(x, y)_B$ )
    | perceptA $\leftarrow$ $\Phi(\Upsilon$(colorPatchA))
    | perceptB $\leftarrow$ $\Phi(\Upsilon$(colorPatchB))
    | errors[$\lambda$] $\leftarrow$ PerceptualError(perceptA, $(x, y)_A$, perceptB, $(x, y)_B$ )
  | **return** errors

**Function** `FindMinimumPrimaries(`*base_errors*`)`:
  | num_primaries $\leftarrow 0$
  | **repeat**
    | num_primaries $\leftarrow$ num_primaries + 1
    | all_tests_passed $\leftarrow$ true
    | trial $\leftarrow 0$
    | **repeat**
      | trial $\leftarrow$ trial + 1
      | Randomly initialize primaries $p = \{p_1, ..., p_{\text{num\_primaries}}\}$
      | **foreach** *wavelength $\lambda$ in WAVELENGTHS* **do**
        | Zero initialize coefficients $\alpha = \{\alpha_1, ..., \alpha_{\text{num\_primaries}}\}$ for primary $p$
        | **repeat**
          | $(x, y)_A$, $(x, y)_B$ = RandomLocations()
          | positive_alpha, negative_alpha $\leftarrow \alpha_i^+$ for all $i$, $(-\alpha_i)^+$ for all $i$
          | testColorPatch $\leftarrow$ MonochromaticColorPatch($\lambda$, $(x, y)_A$ +
            WeightedColorPatch(negative_alpha, $p$, $(x, y)_A$)
          | matchColorPatch $\leftarrow$ WeightedColorPatch(positive_alpha, $p$, $(x, y)_B$)
          | testPercept $\leftarrow \Phi(\Upsilon$(testColorPatch))
          | matchPercept $\leftarrow \Phi(\Upsilon$(matchColorPatch))
          | errors[$\lambda$] $\leftarrow$ PerceptualError(testPercept, $(x, y)_A$, matchPercept, $(x, y)_B$ )
          | coefficients $\alpha \leftarrow$ GradientDescent($p$, $\alpha$, target, match, errors)
        | **until** *convergence*
        | **if** *error $\geq$ base_errors[$\lambda$]* **then**
          | all_tests_passed $\leftarrow$ false
          | break
    | **until** *trial = MAX_TRIALS*
  | **until** *all_tests_passed*
  | **return** num_primaries

---

spectral functions are identical. This "base perceptual error" is larger than the energy of each color percept itself, meaning that the emergent vision fails to recognize a patch as the same color across different parts of the retina.

The second baseline model is a variant of the first, where we change the optic nerve stream to directly transfer cone activation values, omitting the encoding complications of foveated warping and lateral

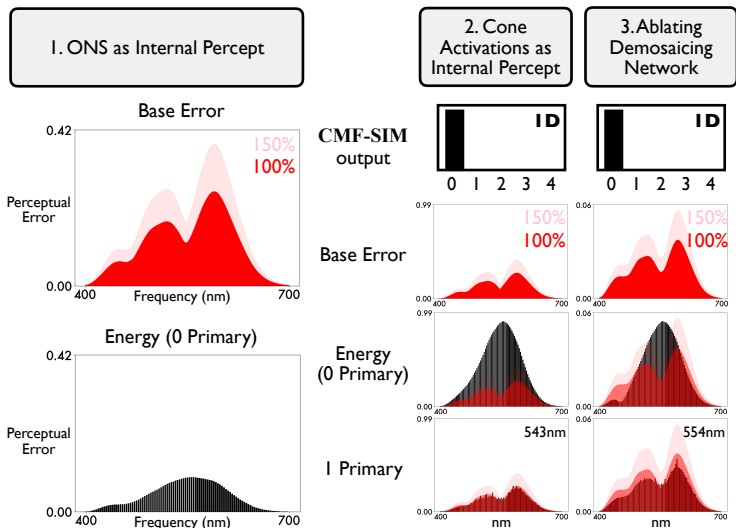

Figure 8: Comparison $CMF\text{-}SIM$ results of our baseline cortical models with limited or no inferential processing. In all of these models, the input optic nerve stream comes from a retina with 3 cone cells as in regular human trichromacy. 1. The first baseline model represents no cortical learning involvement in human visual perception, by setting the internal percept directly equal to the input optic nerve signal. The resulting base error, the perceptual error between the same color patch at two different retinal locations, results in higher errors, compared to the energy of the internal percept. This means that this cortical model fails to recognize the same color consistently across different patches of the retina, so it is procedurally impossible to even try color matching function tests against reference match colors. Formally then, this model fails resolve color vision. 2. In the second baseline model, we treat cone cell activations as internal percepts, omitting lateral inhibition from the eye simulation; $CMF\text{-}SIM$ measures the resulting internal percept as 1D color vision, failing to resolve trichromacy. 3. The third baseline model is an ablation model in which we omit the demosaicking network D from our proposed model of Section B; $CMF\text{-}SIM$ also measures the resulting internal percept as 1D color. Red base errors are superimposed on energy / 1 primary error plots for visual comparison, and 150% of these base errors are applied as threshold determination.

inhibition. This is intended as a far simpler encoding, to pressure test whether color vision can be detected in the spectrally encoded cone values without further processing. However, $CMF\text{-}SIM$ formally measures that this model results in 1D color (Figure 8.2), instead of the expected 3D color from such a retina. This result shows that cortical processing is required even on cone activations to produce color vision of the correct color dimensionality.

The third baseline model adds another perspective by taking the full-featured model detailed in Section B but omitting the demosaicking network $\phi_D$ (as defined in Section B.2.2). As shown in Figure 8.3, $CMF\text{-}SIM$ measures the resulting percepts of this model also as 1D, providing evidence of the importance of demosaicking process in the emergence of color vision.

## E    VARIATIONS IN CELL EXPRESSION AFTER GENE THERAPY

As described in Section 6, we simulate experiments aimed at boosting color dimensionality Mancuso et al. (2009). In our simulation, approximately 60% of M cones are affected by gene therapy, with three possible scenarios for their modification. First, affected M cones are completely transformed into pure L cones (Figure 9.1). Second, affected M cones equally express M and L opsins (Figure 9.2). Third, affected M cones express M and L opsins in random ratios, that is $\alpha L + \beta M$, where $\alpha$ and $\beta$ spatially vary across the retina (Figure 9.3). In all cases, we confirm that our cortical model acquires 3D color vision after adaptation (i.e. re-learning), formally measured by $CMF\text{-}SIM$. This makes intuitive sense if we consider the emergent vision from linear systems theory. Each different opsin can be interpreted as a basis vector for the resulting linear color space. In this view, cells containing a

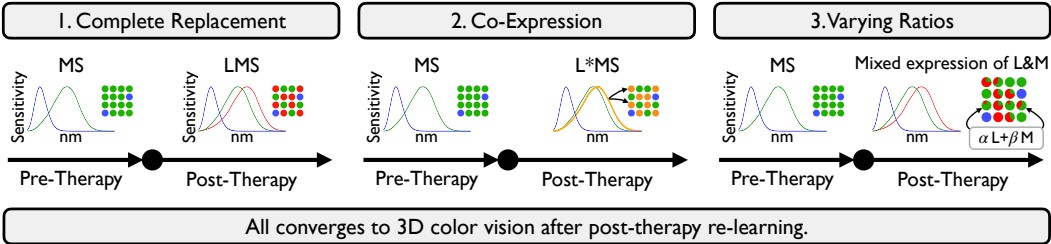

Figure 9: Three scenarios of cell expression after gene therapy adding a third cone type: 1. affected M cones transform into L cones, 2. affected M cones express 50:50 M and L opsins, and 3. affected M cones express M and L opsins at spatially-varying, random ratios (i.e. $\alpha, \beta \in [0,1], \alpha + \beta = 1$). $CMF\text{-}SIM$ shows that all cases converge to 3D color vision after post-therapy re-learning.

mixture of two cones (M and L in this case) can be interpreted as linear combinations of two basis vectors, which cannot increase the dimensionality of the linear space.

## F   PROBING EFFECT OF SPECTRAL RESPONSE AND ENVIRONMENTAL COLORS IN ACQUIRING TETRACHROMATIC VISION

In Section 6, tests of tetrachromatic eyes assumed the fourth cone type was a pigeon Q cone, with peak sensitivity at 506 nm, between the S and M cone peak sensitivities (419 nm and 530 nm, respectively). In this section, we report additional experiments using instead a human Q cone, modeled with peak sensitivity at 545 nm between M and L peak sensitivities (530 nm and 560 nm, respectively). As shown in the spectral graphs of Figure 10, the pigeon Q cone is more decorrelated from L, M, S, which in principle may more easily support emergence of 4D color vision.

Further, we probe the effect of visual environment on the emergence of tetrachromacy, simulating the learning of color vision with three different input scene image datasets. The first environment is the set of real hyperspectral photographs of everyday scenes (Arad et al., 2022) as described in Section A.1. However, natural hyperspectral images are thought to be lacking in human tetrachromatic colors (Lee et al., 2024), so in the second and third datasets we augment the 900 hyperspectral images with 100 tetrachromatic color patches. In the second dataset, the colors are sampled from the tetrachromatic hue sphere as computed using recently developed $N$-dimensional color theory for tetrachromacy (Lee et al., 2024), customized to each of the observers here (i.e. with human Q or pigeon Q, respectively). In the third dataset, the colors are sampled from an idealized tetrachromatic color space in which each of the four cone channels is allowed to take any value, allowing theoretically maximum levels of chromatic contrast. Since natural hyperspectral images have been found lacking in human tetrachromatic colors (Lee et al., 2024), in principle we may expect the likelihood that 4D color vision emerges to increase across these three simulated environments.

Indeed, in line with theoretical intuition, Figure 10 shows that 4D color vision emerges, as measured formally by $CMF\text{-}SIM$, only for the third visual environment in the case of the human Q cone, but 4D color vision emerges with any of the visual environments for the pigeon Q cone. Our simulations suggest strong genetic and environmental effects on emergence of tetrachromatic color vision.

## G   FURTHER EVALUATION OF COLOR EMERGENCE IN CORTICAL MODEL

### G.1   VALIDATION OF SIMULATED COLOR MATCHING FUNCTIONS AGAINST HUMAN PSYCHOPHYSICAL DATA

Our simulation produces results that are highly consistent with the psychophysical measurements of actual human subjects. Specifically, we compared the output of $CMF\text{-}SIM$ with the empirical color matching function data reported in Stiles & Burch (1955). To match the experimental setup in Stiles & Burch (1955), we employed the same monochromatic color primaries at 444nm, 526nm, and 645nm. As shown in Figure 11, the CMFs resulting from color matching simulation in $CMF\text{-}SIM$

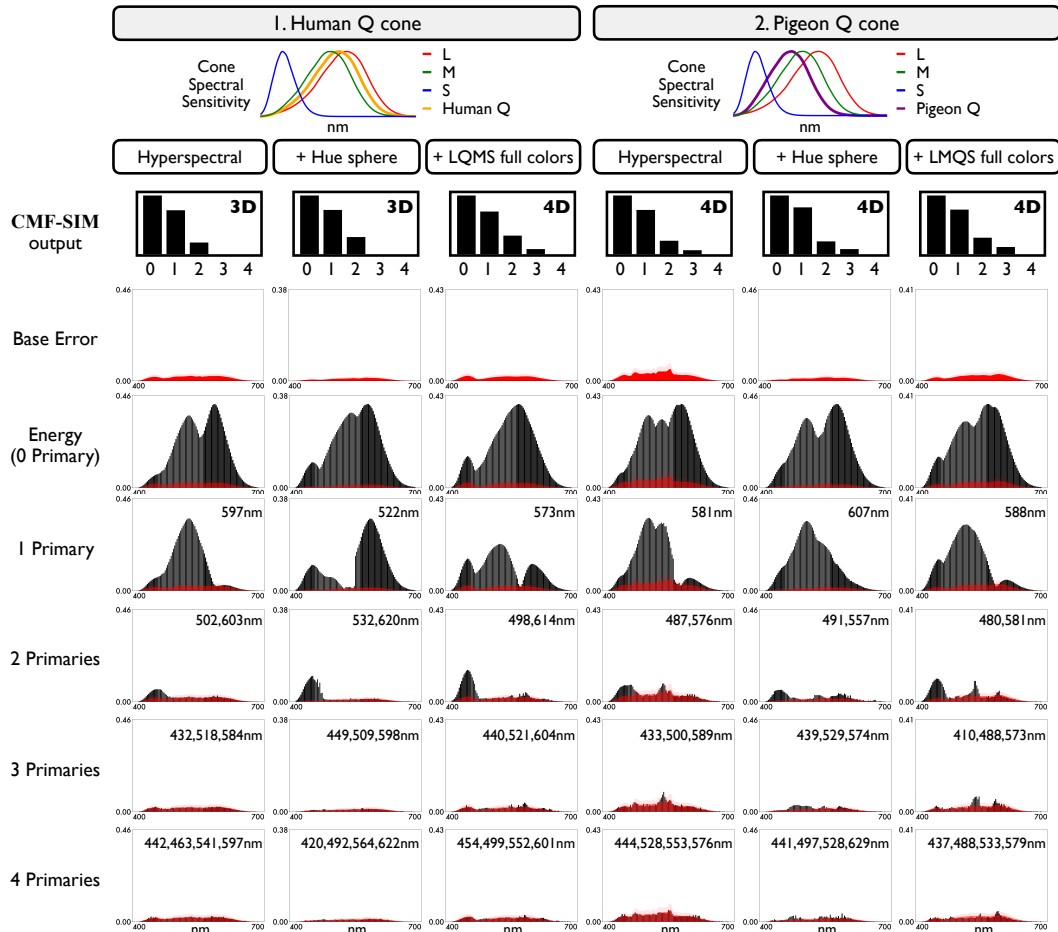

Figure 10: Results of expanded experiments of boosting color dimensionality from 3D to 4D, by addition of a Q cone to trichromatic L, M, S cones. Panel 1 models addition of a Q cone from natural human tetrachromacy, and panel 2 is for a Q cone from pigeon vision; spectral responses are shown. Each panel compares emergence of color vision with three different visual environments: real hyperspectal images (Arad et al., 2022); synthetic images containing tetrachromatic colors sampled from the hue sphere customized to that color observer (Lee et al., 2024); synthetic images containing idealized tetrachromatic colors with chromatic contrast between cone channels at the theoretical maximum. The simulation results show that 4D color vision only emerges for the human Q cone with the third environment, while it emerges for the pigeon Q cone under all three environments.

closely align with the empirical color matching function curves, providing further validation of our simulated cortical model as an accurate representation of human trichromatic color vision.

## G.2 ROBUST EMERGENCE OF CORRECT COLOR DIMENSIONALITY IN CORTICAL MODEL

The emergence of correct color dimensionality in our simulations remained consistent across variations in initialization, designed to mimic the natural biological variability between humans (Carroll et al., 2002; Hofer et al., 2005). We tested this rigorously by varying both cone type ratios and initial noise parameters. First, we altered the L:M cone ratios in the retinal patch, testing 2:1 (original), 1:1, and 1:2, as well as an equal L:M:S ratio of 1:1:1. In all cases, the framework consistently converged to 3D color vision, demonstrating adaptability to different cone distributions. Second, we sampled the random seed for noise parameters, including: photon-shot noise during photoreceptor activation, lateral inhibition noise, Perlin noise for cell position randomization, and cortical model parameter initialization. Despite these changes, the model always converged to 3D color vision when trained

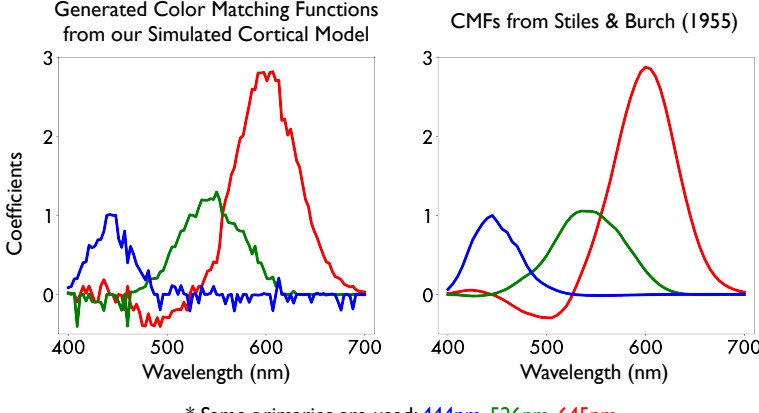

Figure 11: Side-by-side comparisons of two color-matching functions (CMFs) are presented. Left: The CMFs generated from our simulated cortical model, trained using a trichromatic retina. Right: The CMFs obtained from Stiles & Burch (1955), based on real human subjects. To ensure a direct comparison, our simulation used the same set of spectral primaries (i.e., 444nm, 526nm, and 645nm). This comparison highlights the validity of our trained cortical model as an accurate representation of a color observer.

with a trichromat retina, underscoring the robustness of the simulated cortical learning to biological and physical variability.

To further illustrate the robustness of our cortical model, we present its performance across a diverse range of input stimuli during testing, as shown in Figure 12. The figure demonstrates the Neural-Scoped internal percept of our cortical model trained with a trichromatic retina. Specifically:

1. For hyperspectral images derived from RGB inputs, converted using the method described in Section A.1, the model successfully reconstructs accurate color percepts (3rd column) from optic nerve signals (2nd column), which have been time-averaged for improved visualization. Additionally, the model demonstrates complete robustness to various hue variations, accurately reconstructing color percepts even under significant changes in the chromatic properties of the input stimuli.

2. For hyperspectral images from a standard dataset (Arad et al., 2022), the model accurately reconstructs color percepts (6th column) from optic nerve signals (5th column). To aid clarity, we include RGB-projected versions of the hyperspectral images in the 4th column.

This highlights the model's ability to generalize across varying input types and maintain robust performance under diverse conditions.

## H    TABLE OF IMAGES WITH VARYING NUMBER OF CONES FOR PHOTORECEPTOR ACTIVATIONS AND RGC SPIKES

To enhance intuition and visual clarity, we present a table in Figure 13, where the number of cone types in the simulation varies across columns. The rows display images of photoreceptor activations, bipolar signals, and optic nerve signals.

The second row in Figure 13 reveals that photoreceptor activations become increasingly noisy as additional cone types are added to the retinal mosaic. The third row shows bipolar signals, computed by applying a center-surround lateral inhibition kernel to the photoreceptor activations, introducing greater complexity compared to the photoreceptor layer. By the time optic nerve signals (spikes) are generated, differences between cone mosaics become almost indistinguishable, underscoring the cortical challenge of extracting color vision with the correct dimensionality.

We also present a variant of Figure 13 that features the balloon image in Figure 14.

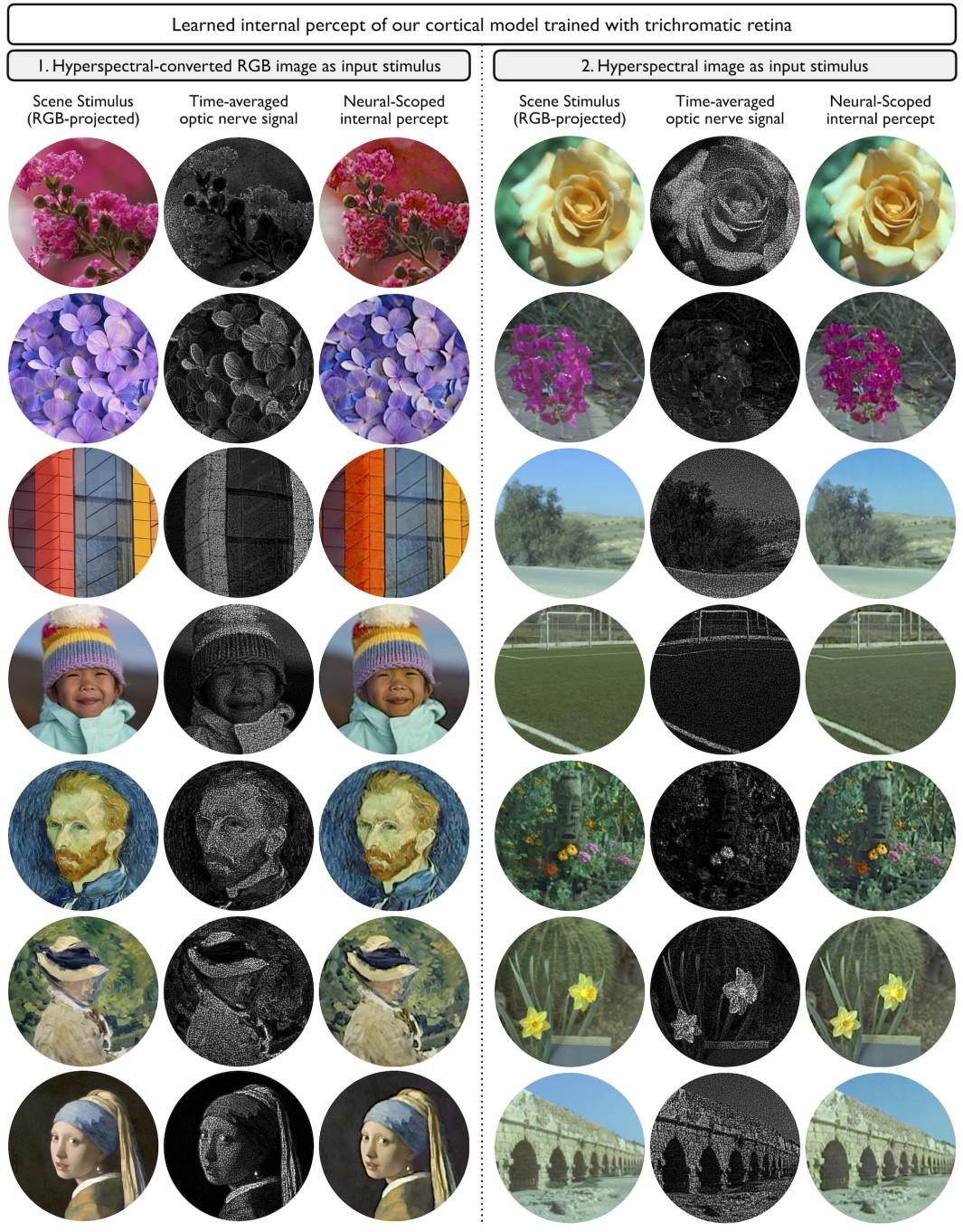

Figure 12: Our learned cortical model demonstrates robust performance across a wide variety of inputs during testing. Here, we present the neural-scoped internal percept of our cortical model trained with a trichromatic retina: 1. For hyperspectral images derived from RGB inputs (Section A.1), the model accurately reconstructs the color percept (3rd column) from optic nerve signals (2nd column, time-averaged for clearer visualization). 2. For hyperspectral images (Arad et al., 2022), the model similarly produces accurate color percepts (6th column) from optic nerve signals (5th column). For visual clarity, we show RGB-projected hyperspectral images in the 4th column.

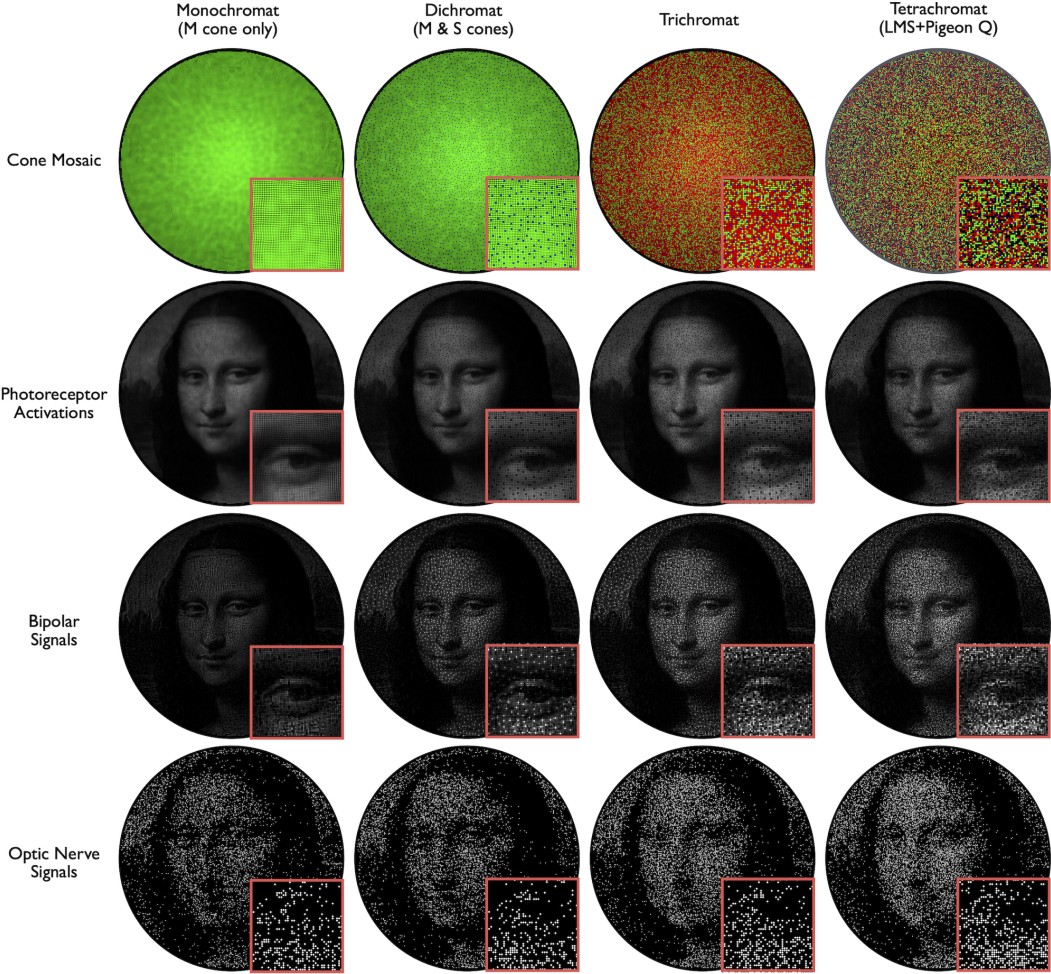

Figure 13: Expanded version of Figure 2.4, showing the full field of view with simulated retinal ganglion cell outputs (i.e., spiking optic nerve signals in the last row). Red boxes highlight close-up details of the full-sized data near the left eye of the Mona Lisa. L, M, S, and Q cones are visualized as red, green, blue, and black, respectively, in the first row.

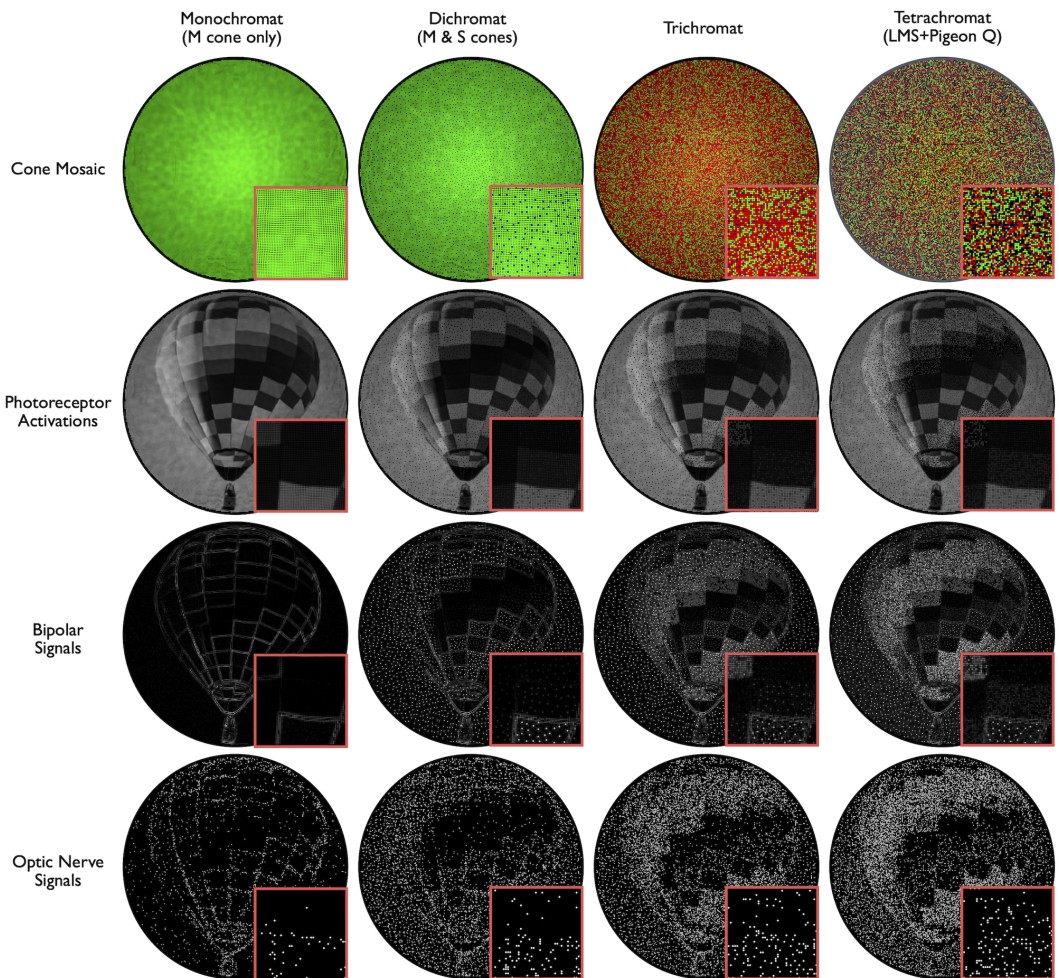

Figure 14: Expanded version of Figure 2.4, a variant of Figure 13, featuring the balloon image.

