# OpenReview forum: "A Computational Framework for Modeling Emergence of Color Vision in the Human Brain"
_ICLR.cc/2025/Conference — ICLR 2025 Oral_

### Official Review · Reviewer_c2wB · 2024-10-31

**Soundness:** 2
**Presentation:** 3
**Contribution:** 2
**Rating:** 8
**Confidence:** 3

**Summary:**

This paper presents a computational framework to explore how color perception might emerge in the cortex through a self-supervised learning model inspired by biological vision. The authors aim to simulate the early visual processing stages by building a model that integrates a retinal layer with different types of cone cells (sensitive to various wavelengths) and a cortical layer using predictive coding. The model learns to extract color and shape information from visual inputs without explicit color labels, reflecting a form of emergent perception.

A significant contribution is the model's ability to simulate the transition from dichromatic to trichromatic vision, akin to genetic interventions shown in experimental biology. The framework demonstrates that under different cone configurations, the model can adapt to various perceptual dimensions, which provides insights into color processing mechanisms in the brain. This interdisciplinary approach contributes to our understanding of self-learning in color perception and offers a basis for future studies on adaptive vision systems in neuroscience and AI.

**Strengths:**

1. The originality stems from the integration of self-supervised learning with a computational framework inspired by biological color perception. By combining a retinal model with a cortical learning mechanism, the authors attempt to create a biologically plausible framework for emergent color and shape perception. This interdisciplinary approach bridges machine learning with neurobiological modeling, which is a valuable exploration in color perception studies.
2 The paper’s methodology reflects a reasonable approach to simulating early visual processing stages. The authors carefully design the retinal model to simulate color cone responses and introduce predictive coding for the cortical learning mechanism. Though some limitations exist, the study provides valuable insights into the use of predictive coding in visual processing models, highlighting how cortical networks could feasibly learn color information.
3 The paper is generally well-written, providing a clear overview of the methods, experiment design, and results. Figures and visualizations are well-crafted, aiding in understanding the experimental findings. The authors successfully explain complex processes, such as predictive coding and color perception, in an accessible way.
4 This study is significant within the niche area of computational neuroscience focusing on color perception. It addresses the unsolved question of how color perception might emerge in a self-learning cortical network, which is a meaningful contribution to understanding visual processing mechanisms. The attempt to simulate gene therapy’s effect on color perception (such as enhancing dichromatic to trichromatic vision) offers a unique perspective that could inspire further studies in perceptual enhancement.

**Weaknesses:**

Biological Plausibility of Input Data:
The model uses natural RGB images as input, which already contain color information. This approach diverges from biological realism, as the retina does not process pre-encoded color data but rather interprets light of various wavelengths. To improve biological plausibility, the model could use spectrally-encoded images or raw spectral data as inputs to better reflect how retinal cones process different wavelengths. This adjustment would align the model closer to real visual processing and allow more meaningful exploration of emergent color dimensions.

Validation against Biological Data:
While the paper presents a computational framework, it lacks direct validation with biological data, which is essential to support its claims of modeling emergent color perception. Adding quantitative comparisons with known biological data on cortical color responses, such as V1 or V4 neuron responses in primates, would strengthen the study’s credibility. Incorporating benchmarks like psychophysical data on color matching could also make the findings more robust and biologically grounded.

Ambiguity in Learning Mechanisms:
The model’s reliance on predictive coding for learning color dimensions in the cortex is interesting but raises questions about biological accuracy, as it assumes self-supervised learning can fully capture color processing. However, in biological systems, factors like feedback from other cortical regions and adaptive changes play significant roles. Expanding the model to incorporate feedback loops or other cortical interactions could offer a more comprehensive view and make the learning process more aligned with biological color perception theories.

Overstated Novelty in Emergent Color Dimensions:
While the paper claims novelty in emergent color dimensions, related work on unsupervised learning of color and perceptual features is well-established. For example, studies on hierarchical models of color vision (e.g., Komatsu, 2006; Conway et al., 2023) and other unsupervised learning models in color processing (Zhou & Mel, 2008) address similar concepts. The authors could enhance clarity by explicitly discussing these foundational works, explaining how their approach extends or diverges from these models, and adjusting novelty claims accordingly.

Komatsu H. The neural mechanisms of perceptual filling-in. Nature reviews neuroscience. 2006 Mar 1;7(3):220-31.
Zhou C, Mel BW. Cue combination and color edge detection in natural scenes. Journal of vision. 2008 Apr 1;8(4):4-.
Conway BR, Malik-Moraleda S, Gibson E. Color appearance and the end of Hering’s Opponent-Colors Theory. Trends in Cognitive Sciences. 2023 Sep 1.

**Questions:**

1 There are many parameters in the model, especially in the cortical learning layers. How were these parameters selected, and are they based on biological references or empirical tuning?
2 The experiments are mostly theoretical, focusing on emergent color dimensions in simplified scenarios. Have the authors considered testing their model on more complex datasets or tasks, such as object recognition with varying lighting conditions?
3  The model relies on self-supervised learning for color perception emergence. Biological systems often use feedback and interaction between cortical areas, which may influence color perception development. How do the authors justify the exclusion of such feedback mechanisms?
4 The paper mentions using divisive normalization for edge detection in the model but does not detail why this specific normalization was chosen over others. Could the authors elaborate on why divisive normalization aligns best with their framework, or if they explored alternatives?

**Details Of Ethics Concerns:**

no concerns

---

> ### Author Response · Authors · 2024-11-19
> **Author Response to Reviewer c2wB (1/3)**
>
> Thank you for your thoughtful review and questions. We present comprehensive answers to each of your questions below.  But we want to begin by thanking you for your detailed description of strengths in the paper, and to summarize our response to the weaknesses you enumerated. First, regarding use of RGB vs spectral images, we need to make an important clarification that all input imagery was indeed spectral (from a dataset of \~900 hyperspectral photographs); this is a critical point, because RGB input imagery would have been a serious flaw in the study.  Second, we heard you ask for validation against real-world data – and we walk through the multiple points of such validation included in the paper.  Third, you asked how we could omit other complexities of brain processing, such as cortical feedback loops – while cortical feedback loops are a valuable future direction, we intentionally designed a parsimonious model to highlight the power of self-supervised learning without relying on such additional inputs. Finally, you mentioned that this research might be considered a niche area, but we wanted to highlight its broader intellectual implications. It tackles a foundational mystery in neuroscience: how perception emerges in the brain from raw sensory inputs. Our hypothesized self-supervised, predictive learning mechanism is generally bio-plausible, and is a promising basis for modeling and studying the emergence of other forms of perception from their sensory organ input streams.  By performing this careful study of this specific problem, we aim to contribute a crucial piece to the larger puzzle of understanding perception and learning.
>
> ### Specific Answers to Weaknesses & Questions
>
> **W1.**  This issue is discussed in General Response B, but this is a very important point and warrants reiteration here.  If the model had indeed used RGB inputs only rather than spectral data, we would consider the study to be fatally flawed. But the answer is simply that input images to the learning simulation were indeed hyperspectral. This fact was annotated in the original manuscript figures and described in Supp A.1, but was not clear enough and confused many reviewers – this is fixed in the proposed manuscript revisions.  We hope that this clarification reassures the reviewer of the integrity of the code, experiments and validation.

---

> ### Author Response · Authors · 2024-11-19
> **Author Response to Reviewer c2wB (2/3)**
>
> **W2.** We agree that validation against empirical biological data is highly valuable, and we emphasize that we do in fact provide several such points of validation in the paper.  We highlight those again here.
>
> First, in line with the reviewer’s suggestion to benchmarking against psychophysical data on color matching, we do present such a comparison in the Supplementary Video at 12:37. It was an omission in the exposition that this was only presented in the video, and we propose as part of General Response C to add this to the supplementary text as well (PR1 Supp G.1).  This comparison is between the raw color matching function values for each test wavelength, between (a) CMF-SIM simulated color matching on the emergent trichromatic color vision (see Fig. 4.2.3) and (b) psychophysical data from real human subjects, as reported by Stiles and Burch \[1\]. The high degree of agreement is an important point of validation against empirical data.
>
> Second, as the reviewer notes, our simulation of gene therapy (Fig.6.1) successfully reproduces the boosting of color dimensionality, consistent with the findings of the gene therapy study by Mancuso et al. \[2\]. In this study, colorblind adult squirrel monkeys were given a third cone type in the retina by gene therapy, and gained trichromatic color vision behavior – they were able to pass color tests that they had failed their entire lives before the gene therapy. Similarly, in our simulation, colorblind dichromat models were given simulated gene therapy by adding a third cone type in the retina, and CMF-SIM simulated color matching testing converges to 3D color – in spite of being tested as 2D color prior to the simulated gene therapy. This comparison to empirical biological study provides another point of validation as the reviewer suggests. In addition, this theoretical work suggests that the monkeys may have gained rainbow-hued vision, as qualitatively assessed by NS color visualization.  This is a point that was not able to be verified in the monkeys, and provides motivation to current biological experiments on attempting to boost color dimensionality in humans in order to test this prediction through conversation with subjects.
>
> Third, we believe it is worth re-emphasizing our core result – showing that 1, 2, and 3 cone types result in 1D, 2D, and 3D color vision. As we noted in response to Reviewer L1yp’s question W4, this result may *seem* like it should obviously work, but we explain why it is a highly non-trivial and novel result. This family of results serves as validation against well-established psychophysical data, including foundational work dating back to Maxwell \[3\] that trichromats require 3 primary colors in color matching tests while colorblind dichromats require only 2\.
>
> Finally, we acknowledge that the reviewer rightly points to additional value in attempting to compare to biological data on cortical color responses e.g. in V1 or V4 of primates – this would be future research.
>
> **W3.** You are absolutely right to point to the complexity of human visual processing, and particularly feedback loops in the brain. It would certainly be interesting to study the present model complexified with feedback loops from other processing centers of the brain. However, we note that such an enhanced model would bring strictly more information into the hypothesized learning mechanism. By this reasoning, the current model is more parsimonious in the inputs it requires and the theoretical implications stronger – the model shows that the hypothesized self-supervised, predictive learning on purely the optic nerve stream can allow color vision to emerge even without explicit feedback loops.
>
> **W4.** We thank you for pointers to these papers. However, and please correct us if we have missed something, but the papers cited do not attempt to tackle the core problem addressed in this paper: how color vision of the correct dimensionality is inferred by the cortex purely from the stream of optic nerve signals. A major contribution of this paper is spotlighting the fact that how the brain achieves this remains a scientific mystery that is commonly sidestepped / overlooked, and to hypothesize an explanation to the mystery.

---

> ### Author Response · Authors · 2024-11-19
> **Author Response to Reviewer c2wB (3/3)**
>
> **Q1.** To respond, let us separate into two sets of model parameters: those in the eye simulation of optic nerve signals, and those in the cortical model where color vision emerges.
> Regarding the model parameters in eye simulation, these are drawn from a textbook vision science model of midget ganglion cell neural circuitry pathways in the retina, and numerical parameters are drawn from the latest available vision science data on this circuit. Specific parameters and citations are detailed in Section 2.1, Section 3 and Supp A.
>
> Regarding the cortical model parameters, these are initialized to neutral, noisy values and learned through the hypothesized self-supervision. In particular, we carefully selected initialization values to ensure that the cortical model did not gain any advantage from prior knowledge of retinal processing. For instance, as shown in Supplementary Video (14:55), the lateral inhibition kernel is initialized as an identity function, cone identities are uniformly distributed, and cell positions are evenly spaced, with no variations in density – all far from the ground truth values. This deliberate setup serves to pressure-test whether the cortical model could independently infer all these retinal properties solely from the ONS stream, intentionally making the inference task more challenging and results more rigorous.
>
> In architecting this set of learnable model parameters, our dual goals were (a) a model that successfully learns color vision from ONS and serves as an existence-proof that the hypothesized learning mechanism works, and (b) was interpretable so that we could assess whether the model had accurately inferred each retinal invariant (e.g. cone identities). Importantly, we make no claim that the brain actually implements this particular engineered pipeline. The focus was on demonstrating any architecture that was learned successfully (non-trivial), and whose internals we could interpret for research insight.
>
> **Q2.** Thank you for raising this interesting point. While it falls outside the current scope of this paper, it represents a promising direction for future work.
>
> **Q3.**  Addressed in response to W3. We reiterate that adding feedback loops from other brain areas would bring strictly more information to the learning mechanism and could accelerate learning, but they are not required. If added to our model they would weaken the core theoretical conclusions of the paper by introducing uncertainty as to whether such feedback loop data is required.
>
> **Q4.** Our apologies, but we did not understand this question. The paper does not “mention using divisive normalization for edge detection in the model.” Could you please clarify the question for us?
>
> In the meantime, we would comment that our paper does not intentionally model edge detection beyond what would functionally result from our implementation of optic nerve signals. Here, our method is to stay as close to biological science as possible, implementing textbook vision science on midget retinal ganglion cell pathways in the fovea, using parameter values from current vision science literature (as detailed in Section 2.1, Section 3 and Supp A). Functionally, edges are amplified by this model due to horizontal-cell-mediated lateral inhibition, as described in Supp A.3.3. Should it be helpful, we also provided an answer related to edge enhancement in response to Reviewer Xmgd’s Q6.
>
> \[1\] Stiles, W.S. and Burch, J.M., 1955\. Interim report to the Commission Internationale de l'Eclairage, Zurich, 1955, on the National Physical Laboratory's investigation of colour-matching (1955). Optica Acta: International Journal of Optics, 2(4), pp.168-181.
> \[2\] Mancuso, K., Hauswirth, W.W., Li, Q., Connor, T.B., Kuchenbecker, J.A., Mauck, M.C., Neitz, J. and Neitz, M., 2009\. Gene therapy for red–green colour blindness in adult primates. Nature, 461(7265), pp.784-787.

---

> ### Comment · Reviewer_c2wB · 2024-11-20
>
> The authors' responses to the comments are largely reasonable and adequately address the concerns raised. Based on the comprehensive explanations provided and the authors' willingness to clarify and validate their approach, I would like to raise the score to 8.

---

> > ### Author Response · Authors · 2024-11-23
> > **Author Response to Reviewer c2wB**
> >
> > Thank you for your thoughtful feedback and for raising the score to 8. We’re glad our responses addressed your concerns and truly appreciate your recognition of our efforts. If there are any other points requiring further clarification, we’re more than happy to elaborate.
> >
> > If our clarifications or revisions have addressed your concerns regarding soundness, presentation, contribution, or confidence, we kindly request you to consider adjusting these scores accordingly. Thank you again for your support.

---

### Official Review · Reviewer_bhn4 · 2024-11-01

**Soundness:** 4
**Presentation:** 3
**Contribution:** 4
**Rating:** 10
**Confidence:** 5

**Summary:**

The manuscript introduces a new approach to computational modelling of colour vision, combining established knowledge of human visual systems with modern artificial neural networks. The proposed framework suggests that the cortex (the agent’s processing unit) lacks direct information about the retina (the agent’s sensory hardware) and investigates the types of colour perception that could arise in such a setup. The model is trained using a biologically realistic principle: predicting future optic nerve signals, aligning with predictive coding theory, a successful approach to unsupervised learning in vision research. To examine the internal representations created by this model, the study employs psychophysical tests commonly used in human colour-matching experiments. The results provide interesting insights into the dimensionality of colour perception at the single-pixel level, offering promising avenues for future exploration into the possibility of tetrachromacy in humans.

**Strengths:**

Overall, I found the article very interesting with several fascinating ideas. Its greatest strength lies in its innovative thinking, presenting a fresh computational approach—sensory-agnostic perception—that stands apart from prior models.

**Weaknesses:**

I would summarise the current weaknesses of the manuscript into three areas:

1. **Presentation**: While generally well-written and easy to follow, some parts of the manuscript require considerable effort from readers to fully understand. For example, abbreviations like "CMF-SIM" and "NS" are not immediately clear. Later, NS is explained as "Neural Scope," but CMF-SIM remains unclear. In the supplementary materials, the "colour matching function" is mentioned, but what about SIM? This ambiguity demands extra effort from readers. Also, there are multiple unresolved cross-references ("??") in the supplementary materials.

2. **Results**: The results section could benefit from being more thorough. Specifically, a broader set of analyses across multiple training instances would strengthen the rigour of the findings and clarify them for readers.

3. **Related Work**: Although the manuscript introduces a unique approach that differs from previous models, a brief section in Related Work could discuss other methods using artificial neural networks to model aspects of human colour perception, such as colour categories, contrast sensitivity in chromatic channels, and colour opponency.

**Questions:**

Below is a list of not only questions but also general comments:

### Optic Nerve Signals
- I think the depiction of optic nerve signals in Figure 1 and the supplementary video should include the idea of distinct photoreceptor cells, which create separate pathways, like parvocellular and magnocellular pathways.
- Following the previous point, in the simulation, if I understand correctly, the optic nerve signal (ONS) is modelled as a 2D matrix, which closely resembles raw images from cameras without further processing. However, since photoreceptor positions are known, a more accurate simulation might use a 3D matrix with the third dimension representing photoreceptor channels. This would mean at each spatial location only one channel contains information (light absorption values). I’d be interested in your thoughts on this.
- In the statement, "[...] colors of the scene are spectrally sampled by different types of color sensitive cells (cone cells) in the retina, appearing as a layer of spatial noise in the ONS [...]", I would argue that raw photoreceptor signals, while noisy, are not spatial noise but spatially meaningful. This is reflected in Figure 3, "True ONS at time t". I think Figure 1 should display a similar visualisation rather than the highly noisy image currently shown.

### Colour Decoding or Encoding?
Throughout the manuscript and in the supplementary video, the authors describe human colour vision as decoding colour. For instance, the abstract begins: "It is a mystery how the brain decodes color vision purely from the optic nerve signals it receives, with a core inferential challenge being how it disentangles internal perception with the correct colour dimensionality from the unknown encoding properties of the eye."

I’d suggest that we do not decode colour vision but rather construct it (in a sense, we encode colour to interpret the world more easily). It is not only from optic nerve signals; our colour perception is heavily influenced by semantic and cognitive processes. I don't mean to play with linguistic terms but raising a conceptual difference. For instance, in my view, the function Phi in Figure 3 "encodes" the internal percept and the CMF-SIM block shows different potential decoding of the internal percept. Currently, Phi is based solely on ONS, but a more complete model would incorporate input from cognitive processes that impact how we encode colour information of a scene.

### Measurement of Colour Dimensionality
I agree that colour is represented in the cortex as a high-dimensional space, $R^N$, but I think $N$ includes more than mono-, di-, tri-, and tetrachromacy. It also encompasses other aspects of colour appearance that are independent of photoreceptor dimensionality, such as transparency, glossiness, etc. In essence, properties of colour perception often relate to objects rather than individual pixels (e.g., see "Colour appearance and the end of Hering’s Opponent-Colors Theory", *Trends in Cognitive Sciences*, 2023, and "Colour perception: objects, constancy, and categories", *Annual Review of Vision Science*, 2018). I’m curious if you’ve analysed different dimensions of the internal percept in the cortical model, as hypothesised, and whether similar concepts emerged there.

### Why Did Human Colour Vision Emerge?
One of the manuscript’s interesting findings is that monochromacy, dichromacy, trichromacy, and tetrachromacy emerge in sequence during the learning steps. This suggests that colour space dimensionality increases evolutionarily without altering the ecological conditions of the system (i.e., the system’s interaction with its input remains the same through the learning process). In contrast, the colour vision dimensionality in biological organisms varies based on their environmental needs. For example, human trichromacy is thought to have evolved for tasks like distinguishing ripe fruit and foliage or recognising emotional states in faces. In the manuscript’s unsupervised learning model, the system’s needs remain constant, yet the colour vision dimensionality expands as learning progresses. I have a few questions about this:
   - What drives this change in dimensionality within the network’s colour vision?
   - How consistent is this emergence? If you repeated the process 100 times, how much variation would appear in the resulting colour vision systems?
   - Why does this evolution stop at four dimensions rather than expanding to five or more?

### Minor Points
- What does a "private-line" visual pathway refer to?
- In Figure 2, for consistency, I suggest using the same balloon image as Figure 1 to clarify the model’s detailed processes.
- In the context of this statement, "[...] random distribution of L, M and S cone cells on the retina with a relative probability of 0.63, 0.32 and 0.05 [...]", it might be helpful to mention the large individual variation in these probabilities, which significantly impacts individual colour perception. Have you experimented with different distributions, and could this framework contribute to understanding this variation?
- In the context of this statement "Specifically our DoG kernel has standard deviations of 0.15 and 0.9 cone diameters, respectively, for the positive center and negative surround Gaussians [...]", the authors might consider modelling dynamic, contrast-dependent centre-surround modulation of lateral inhibition (e.g., see physiological studies by Alessandra Angelucci). This could enhance the efficiency of DoG models (e.g., "Colour constancy beyond the classical receptive field," *PAMI*, 2017, and "Feedback and surround modulated boundary detection," *IJCV*, 2018).

---

> ### Author Response · Authors · 2024-11-19
> **Author Response to Reviewer bhn4 (1/3)**
>
> Thank you for your review and questions.  We are grateful for your encouragement and enthusiasm for our work, and feel that you truly appreciated what we are trying to accomplish. Your detailed comments are thought-provoking and will help us to improve the manuscript, though we have proposed some items as future work below.  Following are answers to each question.
>
> ### Specific Answers to Weaknesses & Questions
>
> **W1.** Addressed in the General Response C.  In sum, we tried to take a thorough pass at improving the exposition and propose specific changes.  We welcome further comments.
>
> **W2.**  Towards addressing this point, we have added a new section, PR1 Supp G, which is referenced in PR1 Section 6\. These are also related to our response to question Q5c below. However, your comment in W2 was quite broad, and if this is not sufficient please let us know what else you have in mind.
>
> **W3.**  We are open to expanding the Related Work to include a section on ANNs as suggested, and have identified a few relevant studies, including work on ANN modeling of color categorization \[1\], contrast sensitivity functions \[2\], and color opponency \[3\].
> However, the description provided is quite broad, and we would appreciate clarification or specific examples to ensure we understood the intended scope effectively. Any additional guidance would help refine this section.
>
> **Optic Nerve Signals**
>
> **Q1a.** The present model is limited to parvocellular midget ganglion cell pathway within 2 degrees of visual angle of the central human fovea. Adding magnocellular pathways (and other of the diverse RGC cell classes) would be future work. One comment is that our intuition was, and the results corroborate, that color vision can in principle emerge purely from the parvocellular pathway data. Therefore, the addition of other pathways is not about finding the minimum conditions required for color vision, but likely about modeling how other types of brain processing (such as fast motion detection), may be assisted or accelerated from RGC classes that compute other derived functions and efficiently communicate the information down the optic nerve.
>
> **Q1b.** This is a clarifying point to consider, and thank you for asking. First, it is indeed true (we have verified) that the cortical model would benefit from representing photoreceptor channels more explicitly by using a 3D matrix representation. However, we intentionally did not model things this way because we argue that it would be a highly unfair assumption and make the problem far easier than reality.  The problem is that such a representation would effectively send the data to the brain on labeled lines for each cone type, relieving the brain completely from having to puzzle out these most challenging unknowns itself. As far as we are aware, there is no scientific evidence that the brain knows the type of each cone a priori, yet it is common to implicitly assume this is true in prior art (discussed in Sec 2.2).  A core contribution of this paper is spotlighting that this seemingly innocuous assumption is actually highly unrealistic, and overlooks the underlying, open scientific question: what are the neural mechanisms that enable the brain to infer not only the type of every cone, but how many cone types there are?
>
> In fact, we observe that the brain clearly adapts dynamically to changes in retinal configurations to maintain its smooth vision. One example is cone death in diseases like retinitis pigmentosa, in which subjects lose function in a large fraction of their cones without visual awareness. This would not be possible if the brain’s photoreceptor labels were static in the cortex, instead of being updated dynamically in adulthood. It is interesting to compare this with a digital camera, in which the demosaicing algorithm leverages the known color identity of each pixel in the color filter array (CFA), such as Bayer filter. However, there could be some malfunctions in some filters in shipment, or some filters simply degrade over time, decreasing the gain levels. Our work shows that each cone spectral identity (or CFA in this example) does not need to be explicitly defined but can be inferred and updated dynamically from self-supervised learning.

---

> ### Author Response · Authors · 2024-11-19
> **Author Response to Reviewer bhn4 (2/3)**
>
> **Q1c.** We fully agree with you that raw photoreceptor signals carry meaningful spatio-chromatic signals, which are the key ingredients from which color vision emerges. We would like to clarify some related points:
>
> There are several distinct types of spatial noise:
>
> - Spiking noise, which dominates Fig.1-ONS
> - Spectral sampling noise with multiple cones (dominates Fig.3.1-True-ONS).
> - Fig.1 shows spiking, but Fig.3 is time-averaged and lacks spiking noise.
>
> These differences were addressed in our supplementary video, but we acknowledge that it was not clearly presented. As Reviewer L1yp Q7 raised an issue of our presentation that some concepts were only explained in the video not text, we propose a new section to our supplementary, as found in PR1 Supp H, that shows a table of images with varying number of cone types and the associated cone activations and spikes.
>
> **Colour Decoding or Encoding?**
>
> **Q2.** We resonate with the idea of interpreting the brain’s function as inferring how to meaningfully encode color from the ONS it receives; this is a valid and insightful perspective. However, it is also rhetorically valid to consider the eye as encoding spectral signals into the ONS –  and therefore it is intuitive to describe what the cortex does in the learned Phi function as decoding what the eye has encoded. While both uses of “encoding” are meaningful, they create a syntactic conflict that would confuse readers if both interpretations are used without precise and careful language. To avoid this potential confusion, we prefer to forgo the suggested interpretation of the brain “encoding” color in this paper, reserving this perspective for future work.
>
> **Measurement of Colour Dimensionality**
>
> **Q3.** Great idea. We have ongoing research studying other scene information (beyond color) that may be inferred by the cortex into the $\\mathbb{R}^N$ pixels of the visual percept. No conclusions yet, but this is a promising direction.
>
> **Why Did Human Colour Vision Emerge?**
>
> **Q4a.** Sequential emergence of color dimensionality is a naturally emergent property of our simulation – it surprised us when we first saw it.  Without any computational biasing, the trichromatic learning progresses through 2D color where the NS images contain only blue-yellow hues (Fig 3.3.1200 steps). We found this remarkable, since blue-yellow hues are known by vision scientists to be the color experience of red-green colorblind dichromats.
>
> Our informal analysis suggests that the cause is due to the relatively large spectral gap between S and M, which seems to make this difference easier to learn. The smaller gap between M and L is also learned, but proves to be numerically more work. We hypothesize that the reason is that L and M signals are more highly correlated, as is well appreciated in vision science \[4\], and therefore take longer to disentangle in the brain's inferred model of the number of different cone types.
>
> Though not included in the paper, further insight comes from simulating variants of the squirrel monkey gene therapy experiment, where the type of the missing cone varies.  In the tritanopia case (M and L cones only, S added), the brain’s mental model has already learned the differences between M and L, so one might guess that the addition of an S cone would resolve to trichromacy more quickly. Indeed, the simulations show this to be true.
>
> **Q4b.** The short answer is that the results are totally consistent. Whether we shuffle the cone mosaic or alter the distribution of different cone types (as discussed in Q7 below), the outcome reliably converges to the same result.
>
> This consistency underscores the primary factors that shape the learned color space. First, scene statistics play a pivotal role. As you might imagine, learning simulation with only grayscale input imagery causes a trichromatic retina model to converge to 1D color vision. Second, cone response statistics – specifically the shapes of cone sensitivity functions (cone fundamentals) – significantly influence the resulting color space. This aligns with the limited color experience observed in individuals with anomalous trichromacy. Together, these factors reveal the interplay between input statistics and biological constraints in shaping color perception.

---

> ### Author Response · Authors · 2024-11-19
> **Author Response to Reviewer bhn4 (3/3)**
>
> **Q4c.**  Actually, we predict that our model will demonstrate emergence of color dimensionality higher than 4 as the number of cone types continues to increase. However, this is an informed prediction, and has not been extensively tested yet mainly because CMF-SIM does not trivially extend to those dimensions due to the computational complexity of the current implementation. Further, our current range of dimensionality (up to 4\) is also backed up by known science – no animals are yet confirmed to have color higher than tetrachromacy and there are likely diminishing perceptual returns in a world with relatively smooth spectral reflectance functions (Jacobs, 2018 \[5\]). For example, scientists do not believe mantis shrimp have ultra-high dimensional color vision even though they possess ultra-high numbers of different spectral photoreceptors.
>
> **Minor Points**
>
> **Q5a.** We refer to the midget ganglion cell pathway in human central fovea that is a circuit from a single cone to midget bipolar interneuron and to midget ganglion cells. In other words, a midget private-line refers to one-to-one mapping from cones to RGCs, unlike outside the fovea where the relationship becomes many-to-one.
>
> **Q5b.** We intentionally chose to use different images here to signal that the model receives different images during training.
>
> **Q5c.** Thank you for the excellent suggestion\! We have included this point in PR1 Section 6 and PR1 Supp G.2, along with relevant citations. Informally, we simulated a wide range of variations, and all configurations consistently converged to 3D color vision. Specifically:
>
> - Original distribution: L:M ratio \= 2:1
> - Variant 1: L:M ratio \= 1:1
> - Variant 2: L:M ratio \= 1:2
> - Variant 3: L:M:S ratio \= 1:1:1
>
> These results suggest that our framework is robust to different cone distributions and highlights its potential for probing the impacts of individual variability on color perception. Exploring this variation further is promising future research.
>
> **Q5d**. That is another great suggestion. Our current model omits effects of temporal cone adaptation and dynamic tuning of lateral inhibition kernels. We strived to build a relatively simple eye model that still captures the intricate mathematical entangling of real visual neural circuitry, to spotlight the significant inferential challenge that the brain must succeed at for color vision to emerge.
>
> From what we have learned so far, our intuition is that further intricate details of real eye encoding (including those you mentioned) are compatible with the simulated emergence of color vision – as long as the brain can represent and tune a mental model of those additional eye encoding functions. The same, hypothesized, self-supervised learning mechanism by sensory prediction can tune those mental model details until it is accurate enough for the brain to see through the encoding to reveal a vision of the world in color.
>
> \[1\] de Vries, J.P., Akbarinia, A., Flachot, A. and Gegenfurtner, K.R., 2022\. Emergent color categorization in a neural network trained for object recognition. Elife, 11, p.e76472.
> \[2\] Akbarinia, Arash, Yaniv Morgenstern, and Karl R. Gegenfurtner. "Contrast sensitivity function in deep networks." Neural Networks 164 (2023): 228-244.
> \[3\] Li, Q., Gomez-Villa, A., Bertalmío, M. and Malo, J., 2022\. Contrast sensitivity functions in autoencoders. Journal of Vision, 22(6), pp.8-8.
> \[4\] Ruderman, D.L., Cronin, T.W. and Chiao, C.C., 1998\. Statistics of cone responses to natural images: implications for visual coding. JOSA A, 15(8), pp.2036-2045.
> \[5\] Jacobs, Gerald H. "Photopigments and the dimensionality of animal color vision." Neuroscience & Biobehavioral Reviews, 2018\.

---

> ### Comment · Reviewer_bhn4 · 2024-11-20
>
> Thank you for your exhaustive effort in preparing the revision.
>
> **W2.**
> Figure 11 in the supplementary material is very interesting.
> - In my opinion, it would be worthwhile to include this figure in the main text, though I understand the constraints imposed by page limits.
> - I would be particularly interested in a brief discussion of your intuitions regarding the small degree of noise observed in the generated colour matching functions.
> - Additionally, in Section G2, the text highlights that results across different initialisations remain consistent. I believe it would strengthen the manuscript to reflect this aspect visually, perhaps by illustrating the variation using shaded regions in the figure.
>
> **W3.**
> All identified papers, along with several others, are relevant to the broader line of investigation pursued in this manuscript. While I do not wish to suggest specific examples, it would benefit your readers to connect this work with a broader body of literature on using artificial neural networks (ANNs) as computational models of colour perception.
>
> **Q1.**
> To a certain extent, I believe the brain has some information about cone types, or at least indirect access to combinations of different cone types. This is supported by findings that the parvocellular layers project to layer 4Cβ, magnocellular layers to layer 4Cα, and koniocellular layers to layer 4A (Callaway, E. M., *Structure and function of parallel pathways in the primate early visual system*, *Journal of Physiology*, 2005). However, I understand the rationale for using a single channel in your framework, given the added complexity of incorporating multiple channels.
>
> **Q2.**
> Fair enough :)
>
> **Q3.**
> I look forward to reading more about this topic in future work.
>
> **Q4.**
> While there is indeed a lack of psychophysical evidence for pentachromacy or higher-order chromatic dimensions in the animal kingdom, computational modelling is a powerful tool to explore such possibilities. Your framework has the potential to reveal the theoretical upper bounds of colour dimensionality, which is a significant strength.
>
> Considering your responses to my comments, as well as those from other reviewers, and the revisions already implemented, I will raise my score to **10**. Congratulations on your excellent work!

---

> > ### Author Response · Authors · 2024-11-23
> > **Author Response to Reviewer bhn4**
> >
> > Thank you for your thoughtful comments and positive feedback. We greatly appreciate the time and effort you have taken to review our work in detail and provide constructive insights. Your comments have been invaluable in shaping our revisions and improving the overall quality of the manuscript.
> >
> > Below, we address some of your points in detail:
> >
> > **W2**: We appreciate your suggestion to include Figure 11 in the main text, as it indeed provides valuable insights. While we will make every effort to integrate it into the final manuscript, this may be subject to page limitations.
> >
> > Regarding the small degree of noise observed in the generated colour matching functions (CMFs), we attribute this to the following factors:
> >
> > 1. *Added Noise in Simulation*: The retina model incorporates various sources of noise (e.g. photoreceptor activation, lateral inhibition), introducing inherent variability in the internal percept. This results in the generated CMFs being slightly noisy rather than perfectly smooth.
> > 2. *Retinal Variability*: As noted in Supplementary Section C, we simulate stimulation across different parts of the retina for test and match color patches. Variability in cone mosaic patterns across regions introduces a small demosaicing error, which manifests as additional noise in the generated CMFs.
> > 3. *Observer Averaging*: In Figure 11, the CMFs represent results from a single trained model. By contrast, the original Stiles & Burch CMFs reflect averages across multiple observers, leading to inherently smoother curves. Informally, we tested averaging our CMF results across multiple trained models (e.g. with different cone ratios) and observed a similar smoothing trend in our simulations.
> >
> > **W3**: Thank you for your suggestion to strengthen the connection between our work and the broader literature on using artificial neural networks (ANNs) as computational models of color perception. While specific citations are constrained by space limitations, we will make an effort to better situate our findings within the wider research context in the final manuscript.
> >
> > **Q1**: We appreciate your insights regarding the brain’s potential access to information about cone types via distinct pathways. The referenced work by Callaway (2005) aligns well with this perspective. We acknowledge the biological plausibility of multi-channel interactions and recognize this as a valuable avenue for future work.
> >
> > In conclusion, we are deeply encouraged by your positive evaluation of our manuscript and thrilled to hear that you have raised your score to 10. Your constructive feedback has greatly enhanced the quality of this work, and we thank you again for your thoughtful engagement.

---

### Official Review · Reviewer_Xmgd · 2024-11-03

**Soundness:** 4
**Presentation:** 4
**Contribution:** 3
**Rating:** 8
**Confidence:** 5

**Summary:**

This paper presents a novel computational framework for understanding the emergence of color vision in the brain, centered on a self-supervised learning model that interprets optic nerve signals without predetermined color dimensions. The framework includes a simulation engine that processes spectral images through realistic eye mechanisms, including eye motion and cone cell sampling, to generate optic nerve signals. The model demonstrates how different types of color vision (monochromatic, dichromatic, trichromatic, or tetrachromatic) can naturally emerge based on the number of cone types in the retina without requiring predefined color encoding. Notably, the framework successfully models enhancement scenarios, such as the conversion from dichromatic to trichromatic vision demonstrated in squirrel monkey gene therapy experiments. It suggests the possibility of enhancing human color vision from trichromatic to tetrachromatic.

**Strengths:**

1. The paper proposes a new model that can simulate the emergence of color vision through a cortical learning mechanism. This model uses self-supervised learning to develop color representation from optic nerve input, demonstrating how color perception emerges naturally from visual data without labeled learning.
2. By simulating color vision without explicit biological pathways, the authors propose a new theory of how color vision could emerge in the brain, suggesting that specific neural pathways may not be strictly necessary for color representation.
3. The model’s insights could inform research into treating or understanding color blindness, possibly providing theoretical guidance for therapeutic approaches.

**Weaknesses:**

1. The paper proposes a cortical learning mechanism for color representation without explicitly mimicking the V1-V2-V4 pathway commonly associated with color processing in biological systems. Does this imply that, in your view, the V1-V2-V4 pathway may not be strictly necessary for achieving color representation? I’m curious if your findings suggest that color perception could emerge through other mechanisms or pathways outside of this traditional hierarchy.
2. Does the model in your paper learn directly from RGB format images? If so, how does it expand from the RGB color space to a 4D color space within the learning process? I’m curious about the mechanisms that allow the model to infer or represent a 4D color space based on RGB input.
3. While your model provides an explanation for the emergence of color representation, I find it challenging to accept that the V1-V2-V4 pathway might not play an essential role in color perception. For instance, the study by Li, Liu, Juusola, and Tang (2014) (see below) on perceptual color mapping in macaque visual area V4 underscores the significance of this pathway in color processing. Could you clarify how your model accounts for or relates to this traditional neural pathway?

> Li, M., Liu, F., Juusola, M., & Tang, S. (2014). Perceptual color map in macaque visual area V4. _Journal of Neuroscience, 34_(1), 202–217. https://doi.org/10.1523/JNEUROSCI.4549-12.2014

**Questions:**

1. In part 1 of Figure 2, fixational eye movements are modeled as small drifts controlled by a uniform probability distribution (*dX*, *dY* ) ∼ *U*{(−15, 15) × (−15, 15)}. I watched the supplementary vedio about the model, it appears (if I'm correct) that the drift pattern displays frequent small adjustments with occasional larger movements. Could you clarify how the uniform probability distribution achieves this effect, given that it should theoretically produce equally likely shifts across the specified range? And why do you choose 15-pixel as the range of movements?
2. I am also curious about the drift pattern moving from the eyes to the mouth. How do the fixational eye movements capture both the eyes and the mouth?
3. I suggest authors review the use of LaTeX \ref{} commands, as there are missing figure numbers in several instances (e.g., lines 71, 141, and 180) in the Supplementary Materials.
4. In your model, could you explain the process used to convert the "scene stimulus" into "incoming spectra"? Specifically, I'm interested in whether you apply any spectral sampling or decomposition steps and if there are transformations based on the eye’s optical properties to simulate the spectra reaching the retina.
5. Could you clarify how cone cell activation is calculated based on the received light? I am wondering whether you apply spectral sensitivity functions for each cone type (L, M, S) or any other specific computational approach to simulate how cones respond to incoming light.
6. ON/OFF pathways in bipolar cells and the center-surround organization in RGCs work together to enhance contrast and edge/contour detection, but in your Figure 2 part 3, I don't see the edge or contour section of the scene has been enhanced?
7. Could you clarify how lateral inhibition works among cone cells in your model? If I understand correctly, this mechanism balances the activation of different cone cells and adjusts their response to light, and then enhances edge and contour. Additionally, could you provide more details on how cone cells, bipolar cells, and RGCs are connected, especially in terms of how these connections contribute to spike generation? I believe these details would be helpful to discuss.
8. In Part 1 of the simulation engine of biological eyes, are there any specific learning rules or algorithms used to adjust parameters or process visual information?
9. Why do you choose the standard deviations of 0.15 and 0.9 in DoG kernel?
10. Could you elaborate on the mechanisms that contribute to the emergence of color representation in your model? From an intuitive perspective, I’m curious how the self-supervised learning and prediction error minimization guide the model toward developing a stable and structured color space, especially as it expands beyond initial color dimensions.
11. Concerning Questions point 10, the model predicts a 4D color space (LMQS) from 3D RGB color images. Since we don’t have four types of cones, we can’t truly envision or confirm what 4D images would look like. Have you tried using 2D colorful images to predict a 3D color space? This approach could help test and validate the model’s predictions.
12. According to Weaknesses point 3. I'm interested in understanding whether the mechanisms in your model might have implications for curing color blindness, particularly in primates such as squirrel monkeys. If the color representation depends on ventral pathway of V1-V2-V4, do you think the your model could potentially inform approaches to treat or "cure" color blindness?  I’d appreciate any insights you could share more information with me.
13. A typo "investigated" on line 156 should be corrected.
14. I don't see the code releases.

---

> ### Author Response · Authors · 2024-11-19
> **Author Response to Reviewer Xmgd (1/4)**
>
> Thank you for your thoughtful review and questions. We value your feedback and help to strengthen the paper.  Our reading is that you have a clear appreciation of the contributions the paper is shooting for, but the main concern may be uncertainty as to whether this theory is compatible with or contradicts existing knowledge on V1-V2-V4 color processing pathways. Our main point would be that the present theory is orthogonal to existing ventral visual pathway processing of color, particularly the V1-V2-V4. The theories do not contradict each other, they may well be compatible, and the relationship between them would be clarified in future research. The theories are orthogonal because this paper studies the computational principles underlying emergence of color vision in the cortex, rather than their mapping onto specific biological structures in the brain. Further, there are various features in V1-V2-V4 processing that are beyond the scope of the modeling in this paper – e.g. chromatic adaptation / color constancy. An important goal of the answers below is to further clarify the relationship to V1-V2-V4 processing, and hopefully allay concerns and earn your strong support for the paper. Therefore, please let us know if we have misinterpreted your concerns, and we would be grateful for your feedback on this response.
>
> The following are answers to each of the questions posed in your review.
>
> ### Specific Answers to Weaknesses & Questions
>
> **W1.** This paper operates at a theoretical computational level of abstraction that is orthogonal to theory of V1-V2-V4 color pathways. Whether our hypothesized self-supervised learning actually utilizes the V1-V2-V4 pathway is beyond the scope of the paper, though we assume that there is considerable overlap. While the organizational structures in these pathways (e.g., blob-like regions in V1, V2, and V4, hierarchical hue refinement, and hue-specific clustering in V2 \[1\]) are important anatomical results, our focus is orthogonal: identifying a bio-plausible computational objective function that the brain could optimize to achieve color vision of the correct color dimensionality. Being at the computational level, this result is independent of claims about the exact anatomical or physiological organization of the brain. We demonstrate that self-supervised learning of sensory signals alone is sufficient for this purpose – a claim that, to our knowledge, has no prior theoretical precedent. These theoretical possibilities may suggest new avenues of research to V1-V2-V4 experts on how the hypothesized cortical prediction, error calculation, and self-supervised learning may be implemented neurally, and where in the brain they are computed. To provide further context, we also added a new subsection, PR1 Section 2.6, that covers prior work in this area.
>
> **W2.** An important point, and addressed in General Response B. To summarize: we use hyperspectral images, not RGB images, as input imagery to the eye model.
>
> **W3.** We agree that the V1-V2-V4 pathway plays an essential role in color perception. But as addressed in W1 above, the modeling work in this paper does not bear directly on how such anatomical pathways relate to the hypothesized learning mechanism.  More generally, many features of the pathway could layer atop the emergence of color dimensionality – examples include the cited neural clustering of hue in V4, and color constancy.
>
> **Q1.** That is a detailed observation, and your question made us realize this requires clarification. In the Mona Lisa video, the eye motions we model are saccades (manually-authored e.g. from mouth to eyes) in addition to fixational drift (modeled as smaller dx,dy over probability distribution as described). But to be clear, these saccades are included only for pedagogical purposes in the video to illustrate regular eye motion with both saccades and drift. We have clarified this point in PR1 Supp A2.
>
> In the learning simulation, we generate our training data using only fixational drifts. We chose 15-pixels as the range of eye motion in order to simulate movements over multiple cones across different parts of the simulated retina. 15-pixels shift corresponds to roughly 15 cones in fovea where cells are most clustered, but 2-4 cones in the periphery (1 deg eccentricity) in our simulation. In sum, this range is set for a practical learning reason.
>
> **Q2.** Addressed in Q1 above. The sudden shift from the eye to the month of the Mona Lisa is an example of a saccade, which we do not simulate for training our cortical model.
>
> **Q3.** Addressed in General Response C.
>
> **Q4.** Our scene stimulus is a hyperspectral image, so a pixel in a hyperspectral image defines an incoming spectrum. The model does not include optical effects such as eye aberrations.

---

> ### Author Response · Authors · 2024-11-19
> **Author Response to Reviewer Xmgd (2/4)**
>
> **Q5.** For each cell location, the scalar activation at that cell is computed as the dot product between the hyperspectral signal (spectrum) at that location with the cone spectral sensitivity function for that cell’s cone type.
>
> **Q6.** There are two main reasons why edge enhancement (which is present) is partially masked visually. First, in RGC activations, there is high noise due to action potential spiking. Second, even in non-spiking bipolar signals (e.g. Fig 2.3 – bipolar), multiple cone types create spatial noise of activation levels. The reason for this is that, even within uniform colored patches, where one might expect low values after lateral inhibition, the actual result is high-energy speckling across the patch. This occurs if, for example. L and M values differ significantly for that color patch – then, the laterally inhibited values will not tend to zero as if it were monochrome. All this described, the edge detection described by the reviewer is actually present – for example, it can be perceived more clearly by comparing cone activations and bipolar signals for the monochromatic case (left column) in Fig.2.4, where the two sources of noise are absent and the edge enhancement dominates. We also added an image example in PR1 Supp H (Fig.13) in which the edge enhancement effect is particularly clear.
>
> **Q7.** The short answer is that we simulate the midget ganglion cell private-line pathway in 2 degree human fovea. Per textbook vision science \[2\], our model reflects lateral inhibition mediated by horizontal cells via H1 and H2 networks (for details, see the cited papers \[3,4,5\]). Mathematically, the functional result of these biological circuits is commonly modeled as a difference of gaussian convolution over the population of cone cell activations (see details in Supp A.3.3, including citations for parameters from most recent vision science measurements). If we focus on a single cone, activity of each cone is also passed into horizontal cells to inhibit activations of its local neighbors.
>
> Structurally, each cone cell in fovea is connected to two bipolar cells (ON and OFF), each of which is connected to a midget RGC (ON and OFF, respectively). There is a direct private-line mapping from each cone cell to two RGCs (ON and OFF). Note, however, that in the cortical model, as a simplifying first step we recombine ON and OFF channels and time-average them (see Supp B.1). The result is that the image inputs to the learning are essentially laterally inhibited cone activations (i.e bipolar signals).
>
> **Q8.** We set parameters in the retina model based on known vision science data and do not change them when converting hyperspectral images into optic nerve signals. In other words, there are no learnable parameters in the retina model – you can think of it as a data generator for our cortical model. We should comment that this is a model and not a biological ground truth; key phenomena are omitted, and could be added for study in future research – an example is time-varying, non-linear adaptation of a cone’s intensity response function \[6\].
>
> **Q9.** As discussed in Supp A.3.3, we simulate the midget private-line pathway in 2 degree human central fovea, and to realistically simulate the receptive field, we computed the standard deviations based on the tabulated fits in Wool et al. \[7\] of the original recording data \[8\].

---

> ### Author Response · Authors · 2024-11-19
> **Author Response to Reviewer Xmgd (3/4)**
>
> **Q10.** The gist of the intuition is as follows.  The brain receives a sensory stream that entangles information about the world with the encoding properties of the sensory organ (eye here). Our model boils down to the notion that, to accurately predict the minute fluctuations of the sensory stream, the brain needs to hypothesize an accurate model of the sensory organ's encoding properties at the cellular scale. If this mental model of the eye is accurate, it allows the brain to disentangle an accurate image of the world from the sensory stream (the visual percept), to mentally translate that image, and to re-encode it with the mental model of the eye to predict the sensory stream changes accurately.
>
> With this mental model, we can intuit how the hypothesized self-supervised prediction guides the brain's mental model to improve. Imagine that learning has converged accurately, but then a cone cell in the retina changes from L to an M cell. The resulting disconnect between the brain's mental model of this cell and its new state will result in larger errors in predicting fluctuations in its value. For example, imagine the cell lies near the boundary between patches of red and green in the world, and an eye motion moves a red patch onto the cell. The brain erroneously believes the cell to be L, so it predicts that the activation will be high (since L is relatively sensitive to red). But the cell is now actually M in the retina, so the real activation for the cell is low. This prediction error is a surprise to the brain, and mathematically guides the brain to update its mental model of this cell towards the new type, M. This mechanism, playing out at the scale of the full population of cells and including the other eye characteristics such as foveation and lateral inhibition, is what provides the possibility for all the eye characteristics to be learned in parallel, accurately and in detail.
>
> Even though the intuition described above explains why the learning *might* work, it is not clear a priori if it would *actually* work. After all, the textbook vision science model entangles many encoding factors in mathematically complex ways.
>
> One way to think of this paper is that it hypothesizes this simple and bio-plausible learning mechanism, and verifies through detailed simulations of the eye and cortex that color vision of the correct dimensionality does indeed emerge.
>
> **Q11.** We apologize for the misunderstanding – as detailed in General Response B, our input stimuli are not RGB images but rather hyperspectral images. The latter naturally support accurate simulation of cone activations in a retina with 4 cone types, and result in 4D color vision (see Fig. 5.1.4) as formally measured by color matching test CMF-SIM.  You are correct that RGB images do not enable emergence of a 4D colorspace, nor do 2D color images support 3D color vision.
>
> **Q12.** Mancuso et al. \[9\] demonstrated that adding an extra photoreceptor type in squirrel monkeys by gene therapy produced trichromatic color vision behavior. However, whether these monkeys experience the same color perception as trichromats remains a core mystery. Our work suggests that it is possible their vision boosted in color dimension to match that of trichromats – a provocative possibility. A requirement in our modeling is that the hypothesized self-supervised learning mechanism continues in adulthood, rather than being limited to development in childhood.
>
> **Q13.** Addressed in the revised manuscript.
>
> **Q14.** We plan to release the code, trained weights and dataset of simulated optic nerve signals upon acceptance, as promised in the original manuscript.

---

> ### Author Response · Authors · 2024-11-19
> **Author Response to Reviewer Xmgd (4/4)**
>
> \[1\] Liu, Ye, Ming Li, Xian Zhang, Yiliang Lu, Hongliang Gong, Jiapeng Yin, Zheyuan Chen et al. "Hierarchical representation for chromatic processing across macaque V1, V2, and V4." Neuron 108, no. 3 (2020): 538-550.
> \[2\] Rodieck, R.W., 1998\. The first steps in seeing. Sinauer Associates.
> \[3\] Rodieck, R.W., 1965\. Quantitative analysis of cat retinal ganglion cell response to visual stimuli. Vision research, 5(12), pp.583-601.
> \[4\] Dacey, D.M., Lee, B.B., Stafford, D.K., Pokorny, J. and Smith, V.C., 1996\. Horizontal cells of the primate retina: cone specificity without spectral opponency. Science, 271(5249), pp.656-659.
> \[5\] Verweij, J., Hornstein, E.P. and Schnapf, J.L., 2003\. Surround antagonism in macaque cone photoreceptors. Journal of Neuroscience, 23(32), pp.10249-10257.
> \[6\] Angueyra, J.M., Baudin, J., Schwartz, G.W. and Rieke, F., 2022\. Predicting and manipulating cone responses to naturalistic inputs. Journal of Neuroscience, 42(7), pp.1254-1274.
> \[7\] Wool, L.E., Crook, J.D., Troy, J.B., Packer, O.S., Zaidi, Q. and Dacey, D.M., 2018\. Nonselective wiring accounts for red-green opponency in midget ganglion cells of the primate retina. Journal of Neuroscience, 38(6), pp.1520-1540.
> \[8\] Crook, Joanna D., Orin S. Packer, John B. Troy, and Dennis M. Dacey. "Synaptic mechanisms of color and luminance coding: rediscovering the XY-cell dichotomy in primate retinal ganglion cells." In The new visual neurosciences, pp. 123-144. The MIT Press, 2013\.
> \[9\] Mancuso, K., Hauswirth, W.W., Li, Q., Connor, T.B., Kuchenbecker, J.A., Mauck, M.C., Neitz, J. and Neitz, M., 2009\. Gene therapy for red–green colour blindness in adult primates. Nature, 461(7265), pp.784-787.

---

> > ### Comment · Reviewer_Xmgd · 2024-11-23
> > **Response to Author**
> >
> > Thank you for your detailed answer. I appreciate the extensive experiments and evidence you provided in response to my questions. I am curious regarding your model's robustness: Is your model resilient to variations in the training set? Additionally, does the emergence of color in your model depend on the distribution of color space? For instance, while the model performs well with images like the Mona Lisa, I am curious about its effectiveness with other types of images, such as a football field, mountains, rivers, and other natural scenes that may have similar hues.

---

> > > ### Author Response · Authors · 2024-11-23
> > > **Author Response to Reviewer Xmgd**
> > >
> > > Thank you for your response.
> > >
> > > Regarding the dependence of color emergence on the distribution of color space in the training set, there is a distinct relationship between the scene statistics of the data used during the cortical learning phase and the resulting learned color perception. A grayscale world results in 1D learned color space for mono-, di-, tri-, and tetra-chromatic retinas. In contrast, a 3D color space world (e.g., one generated only from RGB images, as described in General Response B, rather than employing the hyperspectral dataset [1]) results in a 3D color space for a tetrachromatic retina, because the color dimensionality of the world is inherently limited to 3D in such case.
> > >
> > > Our thesis here is that if the environment possesses a $K$-dimensional color space (e.g. $K=1$ for a grayscale world) and the retina has $N$ types of photoreceptors, the resulting cortical color dimensionality is the minimum of [$K$, $N$]. While this finding is not explicitly stated in PR1, we believe it is a critical point worth emphasizing. Therefore, we propose to include this insight in the final manuscript update.
> > >
> > > Regarding the comments about Mona Lisa and other examples, we would like to clarify a few points:
> > > 1. The cortical learning simulations are not based on a single image input but instead utilize a dataset of hyperspectral images. We used the largest available dataset, containing 900 hyperspectral photos.
> > > 2. Once the model’s learning has converged, it can process any number of images. The examples shown in the paper – such as the balloon, Mona Lisa, and flower – are illustrative but admittedly represent a limited set of outputs, as noted in the reviewer’s response.
> > > 3. In response to this feedback, we have added a new figure pdf in our supplementary materials zip (**Robustness.pdf**). This figure displays inputs (not just RGB images but also hyperspectral examples in proportion), time-averaged ONS, and NS of the learned percept. The grid format differentiates between RGB and hyperspectral examples, demonstrating that the learned model performs robustly across a wide variety of inputs, beyond the few test cases previously highlighted. We plan to include this figure in our final manuscript update as well.
> > >
> > > Finally, on the general topic of model robustness, we would like to draw attention to our responses to Reviewer c2wB’s Q1 (robustness to variation in retina cone type distribution, a well-documented biological phenomenon in human retinas) and Reviewer bhn4’s Q5c (robustness to variation in numerical initialization conditions). These responses underscore that the hypothesized learning mechanism is computationally robust and succeeds as a proof of concept.
> > >
> > > [1] Arad, B., Timofte, R., Yahel, R., Morag, N., Bernat, A., Cai, Y., Lin, J., Lin, Z., Wang, H., Zhang, Y. and Pfister, H., 2022. Ntire 2022 spectral recovery challenge and data set. In Proceedings of the IEEE/CVF Conference on Computer Vision and Pattern Recognition (pp. 863-881).

---

> ### Comment · Reviewer_Xmgd · 2024-11-24
> **Response to Author**
>
> Thank you for the updated Fig. 14 in the supplementary materials. Upon reviewing the middle rows of images, such as those depicting football fields and plants, I noticed that the hue tones are very similar, and the model seems to have difficulty detecting edges in these cases. This makes me wonder if the model's robustness might be limited for image sets with minimal hue diversity. For instance, in the case of football fields, where the grass primarily reflects a single wavelength (represented as green in our visual system), rather than a broader spectrum of wavelengths, it seems that the model’s performance could rely heavily on the diversity of spectral information present in the input. While your results demonstrate robustness to initialization conditions and cone-type distributions, could the model’s performance be less robust when dealing with scenes that lack a wide variety of spectral wavelengths across the electromagnetic spectrum? I’d appreciate your thoughts on this.

---

> > ### Author Response · Authors · 2024-11-25
> > **Author Response to Reviewer Xmgd**
> >
> > Thank you for your feedback.
> >
> > The short answer to your question is no – the model’s robustness is not limited for image sets with minimal hue diversity. We have temporarily included a new file for you (**Robustness_v2.pdf**) in our supplementary material zip, providing additional examples that demonstrate performance in scenarios with minimal hue diversity.
> >
> > We are not entirely sure why you are concerned that limited hue diversity might pose a problem. Technically, we cannot think of any reason this would happen, for similar reasons that a lack of hue diversity does not cause issues in digital cameras.
> >
> > If desired, please feel free to share more details about the underlying reasoning behind your concern, and we would be happy to provide a more technical response in addition to the practical evidence provided in these supplementary images.

---

> > > ### Comment · Reviewer_Xmgd · 2024-11-25
> > > **Response to Authors**
> > >
> > > Thank you for the quick follow-up experiments addressing my question. I greatly appreciate the additional evidence provided in the updated Robustness_v2.pdf file.
> > >
> > > To clarify the reasoning behind my concern, I was wondering how much the emergence of color perception depends on genetic predispositions versus postnatal experience. Specifically, if an infant were unable to perceive a wide range of the electromagnetic spectrum during early development, would color perception still emerge in such a case? That said, your supplementary experiments have sufficiently addressed my concerns, and I’m now satisfied with the robustness of your model. Based on this, I have decided to raise my score from 5 to 8 (as ICLR does not allow a score of 7).
> > >
> > > Additionally, I would like to offer a suggestion regarding your title. While your work focuses on color vision in the human brain, I wonder if the mechanisms of emergent color perception you propose might also apply to other primates or animals. If this is the case, you might consider revising the title to extend its scope to primate brains or other species, which could broaden the relevance and impact of your findings.
> > >
> > > Congratulations on your excellent work!

---

> > > > ### Author Response · Authors · 2024-11-25
> > > > **Quick Response to Reviewer Xmgd**
> > > >
> > > > Thank you, Reviewer Xmgd, for your feedback and for raising your score – we’re excited about your support and will provide a more detailed response soon.

---

> > > > > ### Author Response · Authors · 2024-11-26
> > > > > **Author Response to Reviewer Xmgd**
> > > > >
> > > > > We are excited and grateful for your support.
> > > > >
> > > > > Regarding genetic predisposition via postnatal experience, we deeply share your interest in this scientific question, which remains open. Two points to consider: First, the case of the squirrel monkey, where trichromatic behavior was observed following gene therapy, suggests that “trichromacy can arise from a single addition of a third cone class and it does not require an early developmental process” [1]. Second, the results in our paper point to the intriguing possibility that color perception can emerge later in life, even when the brain has not previously received higher-dimensional parts of the visible spectrum earlier in development.
> > > > >
> > > > > Thank you also for your suggestion regarding the title! We agree that the mechanisms proposed in the paper extend beyond humans and could apply to other animals (and potentially technological approaches). As we prepare the final proposed draft, we will carefully consider this suggestion. One consideration is that the paper’s primary focus is modeling human trichromacy (e.g., many primates are dichromats), though the mechanisms themselves are broadly general.
> > > > >
> > > > > [1] Mancuso, K., Hauswirth, W.W., Li, Q., Connor, T.B., Kuchenbecker, J.A., Mauck, M.C., Neitz, J. and Neitz, M., 2009. Gene therapy for red–green colour blindness in adult primates. Nature, 461(7265), pp.784-787.

---

### Official Review · Reviewer_L1yp · 2024-11-03

**Soundness:** 3
**Presentation:** 3
**Contribution:** 3
**Rating:** 6
**Confidence:** 4

**Summary:**

The authors propose that realistic (because of being 3-dimensional) encoding of color percepts emerge assuming a certain number of spectral sensors and temporal predictive coding. They explore this idea with different number of spectral sensors (or kinds of cones).
In their proposal, first, they learn the different elements of the encoding function by maximizing the consistency (minimizing the error) between the signal captured at time t+dt and the prediction computed from the signal at time t and the displacement information between saccades. Then, the dimensionality of the color percept is defined as in classical colorimetry by looking for the minimal number of independent sources that need to be changed to make a color match (to make the encoded responses the same).

**Strengths:**

The paper is original in proposing predictive coding as a way to fit the elements of color coding.
Another originality is bringing to the machine learning community a concept from vision science: using a response matching experiment to define the dimensionality of the encoding space for certain concept (in this case, the color percept).
However some of the results obtained from this original view seem obvious (see below).

**Weaknesses:**

* Authors claim the emergence of color vision (e.g. titles of the section 4 and 6), however what they reproduce is just the dimensionality of the color space, or the number of illuminants required for response matching. This dimensionality (even if correct) is not a full description of color vision, and should not be reported as that. For example, the dimension does not explain the positive and negative lobes of spectral sensitivity of spectral sensors [Wiesel&Hubel J.Neurophysiol.66], it does not explain their adaptivity under change of illuminant [Von Kries 1902], nor the nonlinearity of their response [Krauskopf & Gegenfurtner JOSA 92], nor the non-Euclidean nature of subjective distances between colors [MacAdam JOSA 42], to cite a few facts.

* Moreover authors refer that the dimension of color space is 3 when assuming 3 different spectral sensors. That is consistent with the result of classical *symmetric* color matching experiments with colors seen in *isolation*, the paradigm of classical tristimulus colorimetry [Wyszeki & Stiles Wiley 82]. However, when seing colors *in context*, the independent dimensions are not 3 but at least 5: not only brightness, hue and colorfulness, but also lightness (not the same as brightness), and chroma (not the same as colorfulness) [see Fairchild, Color Appearanace Models, Wiley 2013]. Therefore, the chromatic content of the spatial context (not only the number of sensors) is critical for the emergence of this 5-dimensional nature, and this is not mentioned in the work. This 3d limitation may come from training limited to regular (white / daylight) illumination, and testing using symmetric color matching with simple (isolated flat) samples, as in the CIE1931 colorimetry. This limitation should be acknowledged and discussed in the work.

* The authors say that previous literature always assumed the dimensionality of the color space. However, there is literature, based on different principles, that can be used to derive the minimal number of sensors and their spectral sensitivity. For instance, [Singh et al. Proc. 3rd Int’l Workshop on Statis. Comp. Theories of Vision 03, Jimenez & Malo IEEE Trans.Geosci.Rem.Sens.14] develop expressions for the accuracy of the recovery of the reflectance of the surfaces and spectrum of the illuminant depending on the use of spatial information and on the number and spectral sensitivity of the sensors. These represent a different principle (inference of object reflectance with color constancy) that can be used to set the dimensionality of the color encoding in a different way as that proposed by the authors.

* The proposed method gives obvious results at some points and these are contradicted at some other points. An example of obvious result is what is said at the abstract (line 23-25): "When the retina contains N types of color photoreceptors, our simulation shows that N-dimensional color vision naturally emerges". However, a later statement seems to contradict the above: in line 352: "CMF-SIM reveals the color dimensionality is 1-D for observers with 3 cone types."

* Methods as the Color Matching Function Simulation or the Neural Scope are not clearly explained in the paper.

**Questions:**

* Authors claim the emergence of color vision. Please tone down / reformulate this contribution. What you get is the dimensionality of the color space.

* The limitation of the 3-d result should be acknowledged and discussed in the work. (My guess) is that richer training and more general testing (e.g. asymmetric matching) would lead to larger dimensionality.

* Cite alternative principles that can be used to derive the number of necessary sensors to encode spectral (or chromatic) information.

* How do you reconcile these two apparently contradictory sentences? (line 23-25): "When the retina contains N types of color photoreceptors, our simulation shows that N-dimensional color vision naturally emerges". (line 352): "CMF-SIM reveals the color dimensionality is 1-D for observers with 3 cone types."

* The Color Matching Function Simulation in the main paper is only explained in FIg.4 and this may not be enough for the average ICLR reader.

* I missed the explanation of the Neural Scope. Note that the generation of images that simulate the perception of dichromats is not trivial (Brettel et al. JOSA 97, Capilla et al. JOSA 04).

* Please do not refer to supplementary videos to explain concepts (as in line 64 of fig.1 or as in line 201 of the supplementary material). If extra explanations are required, please do it through additional sections in the supplementary material.

* Please define the acronyms the first time they appear. Pay special attention to CMF-SIM and NS. I think CVD is not even defined.

---

> ### Author Response · Authors · 2024-11-19
> **Author Response to Reviewer L1yp (1/2)**
>
> Thank you for your thoughtful review and questions. We value your feedback and help in strengthening the paper.  Before answering each of the specific questions in order below, the following summarizes our main takeaways from your review and our responses.
>
> First, we acknowledge that the reviewer found some of the results on color dimensionality self-contradictory and confusing.  General Response A addresses this issue in depth, but given its importance to confidence in the paper’s integrity, we also wanted to discuss it here. To reiterate General Response A, the only result of 1D color from a retina with 3 cone types was a control experiment where the proposed cortical model was entirely removed – the goal of the control experiment was to show that cortical learning of some form is essential.  If this were not true, that result would have been proof that the study had failed\!  We hope that this simple but important clarification can build back your larger confidence in the paper. We invested significant effort in pursuit of rigor and completeness in the validation.
>
> Second, we thank the reviewer for describing key technical points of originality in the paper. We would further highlight our comprehensive enumeration of contributions in General Response D.
>
> Third, the review found that the paper did not sufficiently acknowledge complexity in color vision beyond dimensionality, and had some trouble with the writing.  Our proposals to ameliorate these concerns with improvements in the paper are detailed below.
>
> ### Specific Answers to Weaknesses & Questions
>
> **W1.** We agree that the model of color vision that emerges in this work does not match all features of human color vision. We propose to add text (PR1. Section 2.6) to acknowledge and describe this greater complexity of human vision, and to demarcate the scope and limits of what the present modeling work targets. Your comments on this addition are welcome.
>
> At the same time, we argue that “emergence of color vision” is a valid and helpful description of what is accomplished in this paper, for two reasons.  First, this term refers to the overall process of the cortex learning how to decode the optic nerve stream into something that resembles our everyday experience of vision – not just the color dimension, but seeing through spatial warping, lateral inhibition and patchy spectral sampling. It is perhaps easy to overlook, but should not be underestimated, the dramatic transformation from input optic nerve signals to emergent color vision imagery in our simulation, resembling what occurs in our everyday visual perception (e.g. we adjusted Fig.5 to juxtapose these inputs and emergent colors).  The second reason is that, while the reviewer accurately describes that color perception has important complexities beyond dimensionality (including chromatic adaptation and perceptual nonuniformity of colorspace), we note that color vision researchers have long agreed that the *most critical* characterization of color experience is dimensionality (added Section 2.5 in related work to review for readers). This crucial insight dates back to Maxwell’s foundations of colorimetry and is the basis for color reproduction in all display and printer technology today. It informs the most rigorous characterization of animal color, e.g. in Jacobs 2018 \[1\].  And it is the root of the diversity in human color vision: e.g. grayscale monochromacy, red-green colorblindness, rainbow-hued trichromacy, and 4-dimensional human tetrachromacy (Jordan et al. 2010 \[2\], Lee et al. 2024 \[3\]).
>
> We hope that further research could extend the theory in this paper to encompass the additional layers of complexity in color perception mentioned by the reader.
>
> **W2.** We have added this in the new text on complexity of color perception proposed in W1.
>
> **W3.** To clarify, we were referring to the common modeling assumption in computational neuroscience that the color representation is hardwired as 3 dimensions in the brain (e.g. the dozen papers cited in Section 2.2). With this clarification in mind, our reading is that the Singh et al. and Jimenez and Malo papers are not directly related here – they study a different issue in which it makes sense to process hyperspectral images, but hyperspectral data are not accessible to the brain (none of the dozen papers model the brain receiving hyperspectral inputs). Please clarify further if we have missed the mark.

---

> ### Author Response · Authors · 2024-11-19
> **Author Response to Reviewer L1yp (2/2)**
>
> **W4.** This is a very important point, and is addressed in depth in General Response A.  In summary: the CMF-SIM measurement of 1D color was for a *control* experiment, showing that the raw optic nerve signal itself (without cortical processing) does not produce trichromacy;  and in contrast, the proposed learning-based perception is measured as 3D color, in line with the reviewer’s expectations.
>
> One additional comment here. As the reviewer writes, it *seems* obvious that N cone types in the retina should produce N-dimensional color vision, but in reality this cannot be taken for granted. A contribution of this paper is to spotlight that it is actually a long-standing, open scientific mystery how N cones in the retina results in N-dimensional color vision in the brain, and another contribution is to reason through the requirements for modeling and testing this phenomenon in a rigorous way. (As an aside, K cones do not always create K-dimensional vision in different species \[1\], and a novel theoretical possibility arising from this work is that in those species the neural N is smaller than K.)
>
> **W5**: This was an editing error in our manuscript, and we have addressed this in detail in General Response C.
>
> **Q1:** Addressed in W1 above.
>
> **Q2:** Addressed in response to W4 above, and in General Response A.
>
> **Q3:** Addressed in W3 above.
>
> **Q4:** Addressed in response to W4 above, and in General Response A.
>
> **Q5:** Addressed in W5 above and General Response C.
>
> **Q6:** This was an accidental omission during editing of the manuscript before submission. Addressed in W5 above and General Response C.
>
> In summary, Neural Scope (NS) is a set of visualization parameters that we use to visualize the N-channel image representation of internal percepts in RGB color space. It is simply an Nx3 matrix that is optimized during the cortical training to minimize the projection error.
>
> We would like to highlight the fact that the color space of dichromats presented in the paper, such as blue-yellow and teal-pink hues in Fig.5, naturally emerged in the learned internal percepts. We are not sure if the reviewer is aware, but these are the natural hue-pairs seen by the relevant types of dichromats (blue-yellow for deutan/protan and teal/pink for tritan).  However, we did nothing to try to bias the cortical learning to reproduce these effects – the correct dichromatic hues emerged naturally, and the consistency with biological colorblindness was a surprising and striking point of validation for us.
>
> **Q7:**  Addressed. Some concepts were explained only in the video (including NS) due to editing errors before submission, and have been corrected as discussed in General Response C.  On consideration, we have included video references where time-varying imagery provides insight to the reader that cannot be captured on the static page (such as the video of time-varying ONS signals cited in Fig.1).
>
> **Q8.** Addressed in manuscript, as discussed in General Response C.
>
> \[1\] Jacobs, Gerald H. "Photopigments and the dimensionality of animal color vision." Neuroscience & Biobehavioral Reviews, 2018\.
> \[2\] Jordan, G., Deeb, S.S., Bosten, J.M. and Mollon, J.D., 2010\. The dimensionality of color vision in carriers of anomalous trichromacy. Journal of vision, 10(8), pp.12-12.
> \[3\] Lee, J., Jennings, N., Srivastava, V. and Ng, R., 2024\. Theory of human tetrachromatic color experience and printing. ACM Transactions on Graphics (TOG), 43(4), pp.1-15.

---

> > ### Author Response · Authors · 2024-11-25
> > **Reminder for Reviewer L1yp**
> >
> > Reviewer L1yp,
> >
> > This is a gentle reminder that the discussion period ends in 42 hours. We are pleased to report that after engaging in the discussion, other reviewers have raised their scores: Reviewer Xmgd from 5 to 8, Reviewer bhn4 from 8 to 10, and Reviewer c2wB from 6 to 8. We responded to your original review on 11/19 and wanted to check in to see if you have any further questions or comments. If you find the proposed revisions and the discussion here helpful in clarifying the paper and/or increasing its value, we kindly request that you comment to that effect and consider raising your score before the deadline. Please let us know if you have any final comments, as we aim to upload the proposed final revision of the PDF tomorrow, 11/26, as promised. Again, the final deadline for your response is in 42 hours. Thank you for your time and thoughtful consideration!

---

> ### Comment · Reviewer_L1yp · 2024-11-28
> **Of course the input to the brain is "hyperspectral"**
>
> I disagree with the response W3: this is plain physics (optics). Do not be confused by the (fancy) name "hyperspectral image": every optical image (spatial distribution of irradiance in the image plane of an optical system -e.g. any camera or eye-) has a spectral content in each spatial point. Every image (and also what we have on the retina and is given to the brain) is, of course, hyperspectral. The only images which are not hyperspectral are the digital images (which do not exist in nature). In fact, it is this huge dimensional input what has to be analyzed by the neural system. And the transform to reduce the many-numbers (say thousands of numbers) in the spectrum into a small N because of the use of just N sensors with different spectral sensitivity is the major (design) decision in the neural circuitry for color vision. In this regard I agree with the authors in which the reduction from (say 1000 numbers) -> N numbers is crucial. Nevertheless, my point is that the mentioned papers [Singh,Jimenez] refer to ways of choosing these N kinds of sensors taking into account the spatio-spectral information (which indeed is the input signal to the brain). those are autoencoder-reconstruction-like ways to design these sensors, which are different from the temporal-consistency way proposed by the authors. This is why I think the authors should cite those prevous works based on a different principle. In fact, i think the most original part of this work is this temporal-consistency principle!

---

> ### Comment · Reviewer_L1yp · 2024-11-28
> **Happy to raise my score from 5 to 6 or 8 (assuming proper acknowledgements are done in the intro)**
>
> As I said from the very beginning, what I think is very interesting in this work is the temporal-consistency principle, particularly because that is different from previously reported principles. In my opinion the intro/discussion would be more balanced if *(1)* authors acknowledge that their "explanation of color vision" is limited to the dimensionality (which is crucial, but not the only point), and *(2)* would cite previous explanations (e.g. autoencoder-like) of the design of the sensors and their spectral sensitivity.
> If the authors include those two changes I am happy raising my score from 5 to 6 or 8 to see the paper published.
>
> In the same vein as point (1), a more accurate (more fair, less vague) title for the paper could be: "A Computational Framework for Modeling Emergence of Tri/Tetra-chromacy in the Human Brain". Note that this is what is emerging here (and not the whole "color vision").

---

> > ### Author Response · Authors · 2024-12-02
> > **Author Response to Reviewer L1yp**
> >
> > Thank you for your response and for clarifying the two points you would like to see addressed to fully support publishing the paper.
> >
> > We note that your recent feedback (11/28) was submitted after the revision deadline (11/27), so we cannot make immediate changes to the manuscript at this stage. However, here is how we will or have attempted to address your two points. If these are acceptable, we invite you to adjust your score accordingly:
> >
> > 1. Yes, we can add citations to Singh and Jimenez as examples of explanations of cone cell spectral sensitivity.
> >
> > 2. Regarding the acknowledgment of limitations in our simulated emergence of color vision, you raised this point in your original review (W1 and Q1), and we already attempted to address it comprehensively in two ways: (a) by making amendments to the manuscript, and (b) in our 11/19 response to your original review’s W1.
> > Since you did not refer to either (a) or (b) in your most recent message, we are wondering if you may have missed them. We respectfully refer you to the new Sections 2.5 and 2.6 in the manuscript and have included the text from our W1 response here for easy reference:
> >
> > > **W1.** We agree that the model of color vision that emerges in this work does not match all features of human color vision. We propose to add text (PR1. Section 2.6) to acknowledge and describe this greater complexity of human vision, and to demarcate the scope and limits of what the present modeling work targets. Your comments on this addition are welcome.
> > >
> > > At the same time, we argue that “emergence of color vision” is a valid and helpful description of what is accomplished in this paper, for two reasons. First, this term refers to the overall process of the cortex learning how to decode the optic nerve stream into something that resembles our everyday experience of vision – not just the color dimension, but seeing through spatial warping, lateral inhibition and patchy spectral sampling. It is perhaps easy to overlook, but should not be underestimated, the dramatic transformation from input optic nerve signals to emergent color vision imagery in our simulation, resembling what occurs in our everyday visual perception (e.g. we adjusted Fig.5 to juxtapose these inputs and emergent colors). The second reason is that, while the reviewer accurately describes that color perception has important complexities beyond dimensionality (including chromatic adaptation and perceptual nonuniformity of colorspace), we note that color vision researchers have long agreed that the most critical characterization of color experience is dimensionality (added Section 2.5 in related work to review for readers). This crucial insight dates back to Maxwell’s foundations of colorimetry and is the basis for color reproduction in all display and printer technology today. It informs the most rigorous characterization of animal color, e.g. in Jacobs 2018 [1]. And it is the root of the diversity in human color vision: e.g. grayscale monochromacy, red-green colorblindness, rainbow-hued trichromacy, and 4-dimensional human tetrachromacy (Jordan et al. 2010 [2], Lee et al. 2024 [3]).
> > >
> > > We hope that further research could extend the theory in this paper to encompass the additional layers of complexity in color perception mentioned by the reader.
> > >
> > >
> > >
> > > [1] Jacobs, Gerald H. "Photopigments and the dimensionality of animal color vision." Neuroscience & Biobehavioral Reviews, 2018.
> > >
> > > [2] Jordan, G., Deeb, S.S., Bosten, J.M. and Mollon, J.D., 2010. The dimensionality of color vision in carriers of anomalous trichromacy. Journal of vision, 10(8), pp.12-12.
> > >
> > > [3] Lee, J., Jennings, N., Srivastava, V. and Ng, R., 2024. Theory of human tetrachromatic color experience and printing. ACM Transactions on Graphics (TOG), 43(4), pp.1-15.

---

### Official Review · Reviewer_1FHw · 2024-11-03

**Soundness:** 3
**Presentation:** 3
**Contribution:** 2
**Rating:** 8
**Confidence:** 3

**Summary:**

The authors claimed that they reverse engineered how color vision may emerge in a biologically inspired cortical learning model. They use simulated optic nerve signals (ONS) as input and trained neural networks to individually approximate three invariant properties of the retina: cell positions, cone spectral type, and magnitudes of lateral inhibition. Their model optimizes the error between the predicted ONS signal at the next sampling step and the real simulated ONS signal. They showed that during the minimization of error between inferred ONS and real ONS, the internal perception of their model gradually adds individual primary colors. To show the aggregation of primary colors, they probe the learned embedding with another separately trained model; they call this separate color matching model CMK-SIM. This CMK-SIM is based on text book colorimetry. They use the observation that CMK-SIM found three primary colors in their trained model (instead of the raw optic nerve signal as the model input) as the success of recovering color vision.

**Strengths:**

The authors claim that the main novelty of the paper is the framing of the emergence of color vision as a learning problem. I agree the machine learning framework includes a couple biologically inspired constraints. I would appreciate some clarifications to better understand the significance of the cortical learning model.

**Weaknesses:**

The technical details of this paper are a bit hard to follow (those missing ?? reference in supplementary do not help). For example, I would appreciate further clarification on how the authors choose their baseline models. The authors merely took the entire cortical learning model out (baseline I) or simplified their ONS signal (Baseline II). I guess color/cone spectral type information is present in the ONS signal? Can the author verify this (although CMK-SIM choose to only use 1 color)?

Now another possibility is that the U-net architecture the authors use for the C and D gets way overfit (for example, [1] showed that U-net can generate natural image from white noise inputs and its overfitting occurs between 2400 and 50K steps. The authors here use 100K steps) to recover 3 colors. Considering how much inductive bias is there in the deep architecture, this large number of iterations makes it unclear *where* exactly the color dimensionality emerges. Although the authors postulated the decoder and encoder into factorizable components. There are no ablation studies showing how different learnable components contribute to the emergence of color vision (or if this is not possible given how intricate these components are, would the authors please clarify?)

[1] Ulyanov et al., Deep Image Prior https://arxiv.org/abs/1711.10925

In fact, recent applications of U-net [2] [3] rarely exceeds 5000 steps and they take advantage of its inductive bias on natural images. My main hesitation to fully agree with the results is that the model seems to operate in the "over-fitting" regime to recover the 3rd color.

[2] Falck et al., A Multi-Resolution Framework for U-Nets with Applications to Hierarchical VAEs NeurIPS 2022

[3] Wang, et al., Learning low-dimensional generalizable natural features from retina using a U-net NeurIPS 2022

**Questions:**

A few additional questions:

1. What is $N$ in the cortical learning model? What would happen if the model directly set $N=3$?

2. Does the $NX3$ matrix (neural scope) hold constant in all the shown examples (like Figure 3.3) or is this matrix refitted every single time given the different internal percepts?

3. Would the authors comment on why it takes additional 99K steps to learn the 3rd color? Would this (#steps) be different if there were only 2 colors?

4. If I read the manuscript correctly, the visual percepts are the output of U-net?

5. U-net (the backbone for demosaicking here) contains substantial inductive bias in its architecture. Is the separate translation operator a model merely a biologically inspired model choice? It seems that the model is trained on pairs of data (the input ONS is from time t and the output ONS is from $t+\Delta t$) instead of sequences. If U-net absorbs the learning of $\Omega$, would it change the performance? I understand eye movements shift ONS, but is it necessary to map it into a separate subfunction?

---

> ### Author Response · Authors · 2024-11-19
> **Author Response to Reviewer 1FHw (1/4)**
>
> Thank you for your thoughtful review and questions.  We value your feedback and help in strengthening the paper. We acknowledge and respect your skepticism and/or lack of full clarity about the paper’s central contributions. In General Response D we provide our enumeration of the six main contributions of the paper. If you can forgive the length, here we repeat and elaborate on these six contributions.
>
> The first contribution is spotlighting a long-standing, open scientific mystery, and showing that research progress can be made if we apply interdisciplinary expertise between vision science and machine learning. The open problem is: how does the brain actually construct color vision of the correct dimensionality from the optic nerve signal stream?  We review how it is common to overlook this problem in vision science, instead assuming that the brain knows a priori which optic nerves carry which colors, when this is actually unknown and even the number of retinal cone types is unknown. It is also easy to take for granted that this is an easy problem, given how effortlessly we experience color vision in everyday life, and perhaps given how similar our own vision looks compared to camera processing that is understood at a detailed engineering level. The first contribution is to unearth and spotlight the fact that actually, the emergence of color vision in the cortex is a highly non-trivial inferential computation that remains a scientific mystery.
>
> The second contribution is to reason through and architect new ground rules – a new research game – for studying this open problem in a scientifically rigorous way.  The ground rules we implicitly proposed contain several interlocking parts. 1\) using biologically-realistic optic nerve signals. 2\) restricting simulations of cortex to learn purely from such optic nerve signals. 3\) requiring a mechanistic simulation of cortex that actually learns to produce an internal color vision that can be analyzed. 4\) requiring rigorous analysis of the emergent internal color percept, and spotlighting that such analysis can be as subtle and challenging as analyzing the (unseeable) vision of another human being. 5\) proposing color dimension as a crucial and surprisingly non-trivial characteristic of the emergent color vision, and showing that it can be rigorously measured by adapting Maxwell's famous color matching tests used in human colorimetry.
>
> The third contribution is showing that one can simulate realistic optic nerve signals that model the naturally complex mathematical entanglement of detailed scene and eye characteristics. We do so by bringing in well-known vision science (e.g. foveal midget retinal ganglion cell circuit diagrams from vision science textbooks) and obtaining the latest technical parameters from current vision science literature.  It should be noted that the paper’s simulations of foveal optic nerve signals (e.g. many shown in the video, and code that we would release upon publication) are a first, also a novel contribution of this paper. But even though these simulations are more realistic and detailed than any we are aware of, these are not the final truth. Rather, we hope to inspire further research into modeling increasingly realistic simulation of biological sensory streams.
>
> The fourth contribution is a scientific hypothesis: existence of a specific cortical learning mechanism – self-supervised prediction of optic nerve stream fluctuations under natural eye motions.  This is a novel, bio-plausible hypothesis for a cortical learning mechanism that connects real vision science / neuroscience to machine learning in a powerful, generalizable way.  Note that, within the rules of the new computational neuroscience game, other researchers may certainly propose and study alternative learning mechanisms – we hope they do\!
>
> The fifth contribution is providing an existence-proof simulation that our hypothesized learning mechanism indeed succeeds at explaining the open scientific mystery: that is, the simulation demonstrates emergent color vision that is rigorously measured to have the correct color dimensionality for a biologically diverse range of eyes.  Here, we implemented a specific cortical learning model that was, according to the rules of the game, strictly limited to working purely from the optic nerve signals (e.g. no access to input hyperspectral or RGB imagery, and no knowledge of how many cone types were in the retina).  Note that though we demonstrate the first model to succeed at this task, there are a multitude of alternative pipelines and architectures that could also succeed at this task. Within the context of these six main contributions, though, optimizing this architecture was not the point \-- it is an existence proof that the proposed learning mechanism does the job.

---

> ### Author Response · Authors · 2024-11-19
> **Author Response to Reviewer 1FHw (2/4)**
>
> The sixth contribution is to show different intellectual communities (particularly vision scientists, neuroscientists and machine learning (ML) researchers) the opportunity for deep interdisciplinary research on such long-standing scientific mysteries.  Such interdisciplinary research can unfold in unusual ways.  For example, in our experience, most vision science experts find our simulation of optic nerve signals natural, but do not find it at all obvious that the cortex could compute the cortical unknowns described in this paper.  Conversely, in our experience, most machine learning experts find it almost obvious that ML would succeed at the learning task, but find it a bit strange or even illogical that the eye would encode visual inputs into the optic nerve stream in the way that it does.  This paper was in some sense born of bringing these differing points of expertise and ignorance together.  We hope that vision scientists take away a new appreciation for the far-reaching computational potential of the cortex (as simulatable through machine learning methods), and that machine learning experts gain a clear, case-study view into the potential for deep scientific impact if they delve into the real details of biological reality.  In this paper, we hope that all gain a new appreciation for the wonderful mysteries underlying the color vision we experience everyday.
>
> These six contributions form the contextual perspective within which we considered the reviewer's comments and questions.  Detailed responses to each question are below.
>
> ### Specific answers to questions
>
> **W1**.
> \* *Clarity of technical details*.  This is addressed in General Response C.  Briefly, missing references were fixed, and some conceptual points included only in the video by accident have been added to supplementary text.
>
> \* *How we chose our baseline models*.  Baseline 1 is a control experiment designed to show that the raw ONS data does not itself result in N-dimensional color vision, and establish that cortical learning is required for the emergence of the correct color dimension.  Baseline 2 strengthens understanding of the need for cortical learning, by showing that even when the ONS is simplified to cone activations (without the encoding complexity of foveation and lateral inhibition), N-dimensional color still fails to emerge without learning. General Response A discussed the criticality of these control experiments further.  “Baseline 3” is an extra comparative study (ablation) to show the crucial role of the demosaicing module in the learning model. Supplementary D contains further details about the design of these “Baseline” models and analysis of the results they produce.
>
> \* *Is color/cone spectral type information present in the ONS signal?*  Cone spectral type information is indeed present in the ONS signal, but it is *latent* and *not immediately available*.  These facts are proved by the control baseline experiments that always result in 1D color, while the learning cortical model results in N-D color when there are N cones in the retina. The reason the spectral type is not immediately available is that ONS pixel values are scalar, and the spectral type is not attached to the pixel or known a priori by the brain. This is a fundamental difference from digital cameras, for example, where each pixel is either red, green or blue, and the color of the pixel is known in the image processing pipeline. In the brain, the cortex does not know the spectral type of the cone associated with each ONS pixel, or even how many cone types there are.  One important contribution of the paper is to spotlight this fact, that the cortex must infer these unknowns and construct a cohesive color percept internally, solely from observing the optic nerve data stream.

---

> ### Author Response · Authors · 2024-11-19
> **Author Response to Reviewer 1FHw (3/4)**
>
> **W2 and W3**. Comments to the various points raised:
>
> \* *Overfitting*.  Cheating by overfitting in the conventional sense is not possible in our setting. The simple reason for this is that our test metric (CMF-SIM) is wholly independent from the learning optimization process (self-supervised minimization of ONS prediction error).  A related point is that all the example images in the paper and the video are test images that were unseen by the model during learning.  Third, what is being learned seems to have been misunderstood as somewhat trivial, which leads to the next comment.
>
> \* *The learning model takes “100K  steps to… recover 3 colors”*.  Described this way, the result seems somewhat trivial and the risk of overfit does seem high. But the reality is very different, and we would like to comment to ensure we are on the same page.  First, what emerges from the learning model is not just 3 colors, but rather a 3-dimensional color space containing thousands of possible colors, within images containing thousands of color pixels. Second, this inferential task is computationally demanding, because the model must learn to generate a smooth, undistorted color image while simultaneously inferring (1) the number of cone types, (2) the spectral identity of each cone, (3) the lateral inhibition kernel, and (4) the spatial position of each cone cell. These complexities, necessitated by the biological nature of human optic nerve signals, make the inference far more challenging than standard inpainting or denoising problems.
>
> A potential point of confusion may be the seemingly small number of color primaries (e.g. 3\) used in CMF-SIM. But this also should not be underestimated. These color matching tests are based on colorimetry fundamentals, established by Maxwell, that an observer’s color vision is N-D if and only if the minimum number of color primaries required to match all possible colors is N.  The difference between the need for 1, 2 and 3 color primaries is a fundamental change in color vision experience – from grayscale monochromacy, to blue-yellow dichromacy (colorblindness), to rainbow-hued “natural” trichromacy (Fig 5.1).  While it may seem obvious that having 1, 2 or 3 cones should confer 1D, 2D or 3D color vision, this is not a given (e.g. see Jacobs 2018 \[1\]). In sum, one of the main contributions of this paper is introduction of the framework for rigorously evaluating the color dimensionality of color vision that emerge from simulations of the brain processing the optic nerve stream.
>
> \* *Ablation studies*.  As discussed in W1 above, our baseline models are ablation studies. However, unlike many computer vision papers, it would not have been meaningful to conduct comprehensive ablations of the architecture. We are not claiming that this architecture is exactly what is occurring in the brain.  Rather, this specific architecture is provided as an existence proof that it is computationally possible for the brain to use the hypothesized self-supervised learning mechanism to infer color vision purely from the unlabeled optic nerve stream data.  Per our introductory comments, the main paper contributions here are to spotlight that the brain needs to do so, that how it does so remains an open scientific mystery, and that the existence proof model shows that the brain could do so through the hypothesized self-supervised learning mechanism.
>
> **Q1**: In the learning model, N represents the dimensionality of the internal percept in the color axis and serves as the upper limit for the potential color dimensionality that the cortical model can achieve. Your understanding is correct – N=3 is sufficient to produce a 3D color percept for a trichromatic subject. However, for a subject with 4 cone types, N=3 on the cortical side would be insufficient to capture the full 4D color information, resulting in emergent color vision that was neurally-limited to 3D color.
>
> **Q2**: Generally we compute a different Neural Scope (NS) matrix for each cortical model instance, and apply this matrix across all output images from that model.  For Fig.3.3 and videos, we compute the NS matrix for each timestep as the model is being learned.  But it is worth re-emphasizing that the NS matrix is computed independently of and has no effect on the hypothesized cortical learning – NS is only a visualization tool.

---

> ### Author Response · Authors · 2024-11-19
> **Author Response to Reviewer 1FHw (4/4)**
>
> **Q3**:  Our interpretation is that the 3rd color dimension takes longer to emerge because the M and L cones are more highly correlated (a well known vision science phenomenon, and see L, M, S plots in Fig.2.2). The number of steps is less if there are only 2 cones, with the M or L cone missing.
>
> To help see this, we have generated a new video that showcases the learning progress in various models with respect to the learning steps (**Learning\_Progress.mp4** in our revised Supplementary Material zip). Two comments:
>
> 1. 100K steps is not a requirement for emergence of the 3rd color dimension. Full color vision does emerge after about 15K steps as you see in this movie. We ran training to 100K to ensure the model had converged (see loss graph in Fig.3.2).
> 2. Evidence that M and L correlation slows emergence of color dimension is visible for the tritanopia model (L and M cones only), compared to the relatively quick deuteranopia model (L and S only) or protanopia (M and S only).
>
> **Q4**: Yes, the output of U-Net, D, is the (warped) visual percept.
>
> **Q5**:  Great question. As you suggested, we did an experiment trying to incorporate module Ω (and P also needs to join) into the U-Net, D.  However, we could not get it to work easily, because the U-Net did not represent the spatial unwarping function of P and Ω accurately.
>
> This said, performance (speed) of our existence-proof cortical architecture was of peripheral importance to us. Our primary focus was on achieving an existence-proof simulation that demonstrated the hypothesized learning mechanism works – successful. Next, we placed value in a model architecture factorized into neural buckets that could be interpreted (hence less insight from combining D, Ω and P.
>
> \[1\] Jacobs, Gerald H. "Photopigments and the dimensionality of animal color vision." Neuroscience & Biobehavioral Reviews, 2018\.

---

> > ### Author Response · Authors · 2024-11-25
> > **Reminder for Reviewer 1FHw**
> >
> > Reviewer 1FHw,
> >
> > This is a gentle reminder that the discussion period ends in 42 hours. We are pleased to report that after engaging in the discussion, other reviewers have raised their scores: Reviewer Xmgd from 5 to 8, Reviewer bhn4 from 8 to 10, and Reviewer c2wB from 6 to 8. We responded to your original review on 11/19 and wanted to check in to see if you have any further questions or comments. If you find the proposed revisions and the discussion here helpful in clarifying the paper and/or increasing its value, we kindly request that you comment to that effect and consider raising your score before the deadline. Please let us know if you have any final comments, as we aim to upload the proposed final revision of the PDF tomorrow, 11/26, as promised. Again, the final deadline for your response is in 42 hours. Thank you for your time and thoughtful consideration!

---

> > ### Comment · Reviewer_1FHw · 2024-11-26
> >
> > I have read through the author's responses. I agree with other reviewers that the authors did a great job addressing the reviewer's questions. I have changed my score to 8.

---

> > > ### Author Response · Authors · 2024-11-26
> > > **Author Response to Reviewer 1FHw**
> > >
> > > Thank you for your feedback and for increasing the score to 8. We’re glad our responses addressed your concerns. If our revisions have resolved your concerns about soundness, presentation, contribution, or confidence, we kindly ask you to update those scores as well. Thanks again for your support.

---

### Author Response · Authors · 2024-11-19
**General Response (1/3)**

We would like to express our sincere appreciation to all reviewers for dedicating their time to reviewing our submission and providing insightful feedback to improve our manuscript.

We have responded to each reviewer’s questions individually.  But as a whole, we found 4 major areas in the review set that we have addressed collectively here in General Responses A-D.  Briefly, A and B clarify two simple but rhetorically crucial points of the paper that were misunderstood by multiple reviewers (due to lack of clarity in the original manuscript). C covers the breadth of writing improvements we have made in response to reviewer feedback, incorporated into a proposed revision to the manuscript. The revised main manuscript is accessible via **the “PDF” button** at the top-right of this page. The revised supplementary material can be found in the updated Supplementary Material zip file, named **Proposed\_Revision\_v1\_Supp.pdf**. D is a summary restatement of the six main contributions of the paper.  Should reviewers have the time, we would be grateful for your further comments and/or questions.

Below, we have shared a schedule of when we expect to deliver our final proposed revision to the manuscript.  Our goal in these revisions is to try to clarify and improve our manuscript as much as possible based on your collective feedback.  We hope to earn your increased support for publication, and if you feel we are successful, we would be grateful for your adjustment of your reviewer score and summary. Thank you for your time and consideration\!

**Proposed Next Steps / Calendar Through This Discussion Period**

11/19 (Tue) - First Proposed Revision (revised manuscript files, hereafter referred to as **PR1** for convenience) by Authors (**attached**)
11/22 (Fri) - Feedback from reviewers (soft deadline for incorporation into final revision)
11/26 (Tue) - Final Revision by Authors, merged into the final_manuscript pdf
11/27 (Wed) - Final update of reviewer scores and summaries
11/28 (Thu) - Discussion period formally closes

**General Response A: Control Experiment (Baseline) was Mistaken for Real Results**

This was an important point where writing in the manuscript was not sufficiently clear, caused serious confusion, and undermined core results in the paper.  We have fixed this in the revised manuscript (Section 6 and Fig.5), and explain what happened here.

We acknowledge that multiple reviewers found some of the results on color dimensionality self-contradictory and confusing.  In particular, how and why would a retina with three cone types result in vision that was measured as only 1D by the proposed color matching function test CMF-SIM? This is an extremely important question – we note that if this result were real, it would be a dagger in the heart of the paper’s central claim that color vision emerges *with the correct dimensionality*.  What happened is simply that the passage in the original manuscript (Line 352\) was unclear – the 1D color from 3 cone types was not a result of the proposed learning model, but rather the result of a control experiment. The control experiment showed that *omitting the entire cortical learning model* reduces color vision from 3D to 1D, and the point was to prove that some form of cortical learning is essential – as proposed.  Indeed, as shown in Fig.5, N-dimensional color vision is correctly learned by the proposed model when there are N cone types.  We hope that clarifying this important misunderstanding can build back reviewer confidence in the paper. We invested significant effort in pursuit of outstanding rigor and comprehensive validation of the claims.

---

> ### Author Response · Authors · 2024-11-19
> **General Response (2/3)**
>
> **General Response B: Input Images to the Eye Model are Hyperspectral, Not RGB**
>
> This was another important point where presentation in the manuscript was not sufficiently clear, caused serious confusion, and undermined confidence in the core validity of the study.  We have clarified this throughout the revised manuscript, and explain what happened here.
>
> Multiple reviewers thought image inputs to the model were RGB rather than hyperspectral.  If this were true, we would ourselves have considered the study to be seriously flawed – RGB input images would lack the spectral dimensionality to model tetrachromats, and also weaken analysis of all other cases by forcing input images to a 3D spectral subspace. Fortunately, this is a simple misunderstanding – the inputs to the model are all hyperspectral images. For simulation of cortical learning, the inputs were the largest available database of real hyperspectral photographs \[1\]. For completeness, we should also explain that an exception to this are the photos in manuscript Figures (Mona Lisa, balloon, flower – selected for compositional clarity at small size) – these were RGB promoted to spectral images by simulating presentation on physical displays with hyperspectral red, green and blue pixels \[2\] (details added in PR1. Supp A.1). The use of hyperspectral images was described in the original manuscript, but we have increased clarity and detail throughout the revised manuscript (PR1. Fig.1 & 2 and in PR1. Supp A.1). We hope that this clarification reassures reviewers as to the integrity of the code, experiments and validation. Again, we invested significant effort in pursuit of outstanding rigor and comprehensive validation of the claims.
>
> **General Response C: Proposed Writing Improvements in Light of Reviewer Feedback**
>
> Thank you to all reviewers for your careful reading and thoughtful feedback on how to improve the writing and presentation.  We have incorporated all your suggestions and attempted to address all deficiencies in the attached, proposed revision for the manuscript.  The main changes include:
>
> * Fixed missing Latex references and missing acronym definitions (sorry)
> * Clarifications and/or additional detail on: control experiment, hyperspectral, CMF-SIM color matching test, saccades, Neural-Scope (NS), signal spatial noise due to spectral sampling vs spiking.  This includes adding some material into the main text that was only included in the video.
> * Clarity and prominence of detail to avoid a false impression of technical triviality or obviousness of core neural inferential problems.
> * Related work additions suggested (e.g. PR1 Section 2.5 & 2.6)

---

> ### Author Response · Authors · 2024-11-19
> **General Response (3/3)**
>
> **General Response D: Restatement of Paper Contributions Following Consideration of Reviews**
>
> In reading the reviews, we were encouraged by the recognition amongst a majority of reviewers for each of the main contributions we were aiming for.  However, we observed that no individual reviewer noted all the contributions. Some of these are intentionally left implicit in the paper, but for the purpose of this discussion we wish to be explicit about what we view as the six main contributions of this submission.  Should it be helpful, these six contributions are also elaborated upon in the response to Reviewer 1FHw below.
>
> 1. Spotlighting the long-standing, open scientific mystery of how the brain constructs color vision of correct color dimensionality, purely from the optic nerve signal stream.
> 2. Reason through and establish the ground rules for studying this open problem in a renewed and rigorous manner. The proposed rules include: using biologically realistic optic nerve signals; cortical model inputs strictly limited to these signals; proposed cortical model must produce an internal color vision that can be analyzed as a black box akin to human observer; analysis to include rigorous measurement of color dimension by Maxwellian color matching. (We annotated Fig.1 to bring out this structure.)
> 3. Showing how to implement realistic simulation of foveal optic nerve signals through careful implementation of the latest vision science literature.
> 4. Scientific hypothesis of a specific cortical learning mechanism: self-supervised prediction of optic nerve stream fluctuations under natural eye motions. This novel, bio-plausible hypothesis connects neuroscience to machine learning in a powerful, generalizable way.
> 5. Providing an existence-proof simulation that the hypothesized learning mechanism succeeds at explaining the open scientific mystery. This simulation demonstrates emergent color vision that is rigorously measured to have the correct color dimensionality for a biologically diverse range of eyes and real experiments.
> 6. Showcase for different intellectual communities (particularly vision science, neuroscience and machine learning) the opportunity for deep interdisciplinary collaboration on a long-standing, open scientific mystery.  The hope is for researchers to catch a glimpse of the profound impact they could have on the other, if they are willing to delve into the real details across the intellectual fence.
>
> We have strengthened presentation of these contributions in the writing and figures throughout the revised manuscript.
>
> \[1\] Arad, B., Timofte, R., Yahel, R., Morag, N., Bernat, A., Cai, Y., Lin, J., Lin, Z., Wang, H., Zhang, Y. and Pfister, H., 2022\. Ntire 2022 spectral recovery challenge and data set. In Proceedings of the IEEE/CVF Conference on Computer Vision and Pattern Recognition (pp. 863-881).
> \[2\] Brainard, D., Jiang, H., Cottaris, N.P., Rieke, F., Chichilnisky, E.J., Farrell, J.E. and Wandell, B.A., 2015, June. ISETBIO: Computational tools for modeling early human vision. In Imaging Systems and Applications (pp. IT4A-4). Optica Publishing Group.

---

### Author Response · Authors · 2024-11-22
**Quick Response**

Reviewer bhn4 and Reviewer c2wB,

Thank you for your support and help improving our paper.

We were thrilled to see your scores raised to 8 and 10.
We hope to earn strong support from the other reviewers as well.

Reviewer bhn4, we will come back with a more detailed reply to your questions later.

---

### Author Response · Authors · 2024-11-27
**Final Manuscript Revision**

We have uploaded the final main manuscript PDF and supplementary material ZIP.
Compared to PR1 (uploaded on 11/19), we have made changes as outlined below:

- All PR1 proposed revisions have been applied and merged into the main text—red and blue highlighted texts have been removed in the new version.
- Reviewer bhn4, regarding Figure 11, we decided not to squeeze it into the main paper. Instead, we have referenced the appendix G1 that contains it more prominently in the main paper (Line 414).
- Appendix G.2 has been updated with a new figure and accompanying text description to further strengthen the robustness evaluation section, addressing feedback from Reviewer Xmgd.
- Figure 1 has been updated with a new image to ensure that foveal distortion is clear at a glance in the ONS.

Thank you again for your review and feedback, which has significantly improved the quality of the manuscript. The final deadline for PDF revisions is approximately 24 hours from now. Please let us know if you have any last-minute feedback or notice anything we may have missed.

---

### Public Comment · ~Atsunobu_Kotani1 · 2025-02-12
**Project Website + Code Release**

We would like to thank the reviewers and area/program chairs for their valuable feedback, and we are thrilled that our paper was accepted as an oral presentation.

As promised, **[full code is released](https://matisse.eecs.berkeley.edu)** with this acceptance, and we are seeking users and feedback to continue developing this research direction.

---

### Meta-Review · Area_Chair_FBtn · 2024-12-11

**Metareview:**

This paper introduces a novel computational framework to model the emergence of color vision in the human brain. Using biologically realistic simulations of optic nerve signals and a self-supervised learning mechanism, the study demonstrates how the brain can infer color dimensionality without predefined assumptions. The work effectively bridges neuroscience and machine learning, addressing a fundamental mystery in vision science. The authors validate their approach through rigorous experiments, showcasing applications to both normal and enhanced vision scenarios. While the paper could benefit from a deeper discussion of perceptual complexities and prior related works, its innovative contributions and interdisciplinary significance strongly support the acceptance of this paper.

**Additional Comments On Reviewer Discussion:**

During the rebuttal period, the authors addressed key concerns from reviewers effectively. They clarified that the primary contribution lies in modeling the emergence of color dimensionality while acknowledging the limits of the work in capturing broader perceptual complexities. Detailed explanations were added for mechanisms like Neural Scope and lateral inhibition, and missing references and acronyms were corrected. They contextualized their work within related literature, highlighting its novelty in using temporal consistency as a principle. Concerns about biological plausibility and overfitting were addressed with clear arguments and evidence of robustness. The revisions improved the paper’s clarity and scope, leading to increased reviewer confidence and supporting the decision to accept.

---

### Decision · Program_Chairs · 2025-01-22

Accept (Oral)